# Discrete Flow Matching

**Itai Gat**[1]     **Tal Remez**[1]     **Neta Shaul**[2]     **Felix Kreuk**[1]     **Ricky T. Q. Chen**[1]

**Gabriel Synnaeve**[1]        **Yossi Adi**[1]        **Yaron Lipman**[1]

[1] Meta FAIR    [2] Weizmann Institute of Science

## Abstract

Despite Flow Matching and diffusion models having emerged as powerful generative paradigms for continuous variables such as images and videos, their application to high-dimensional discrete data, such as language, is still limited. In this work, we present Discrete Flow Matching, a novel discrete flow paradigm designed specifically for generating discrete data. Discrete Flow Matching offers several key contributions: *(i)* it works with a general family of probability paths interpolating between source and target distributions; *(ii)* it allows for a generic formula for sampling from these probability paths using learned posteriors such as the probability denoiser ($x$-prediction) and noise-prediction ($\epsilon$-prediction); *(iii)* practically, focusing on specific probability paths defined with different schedulers improves generative perplexity compared to previous discrete diffusion and flow models; and *(iv)* by scaling Discrete Flow Matching models up to 1.7B parameters, we reach 6.7% Pass@1 and 13.4% Pass@10 on HumanEval and 6.7% Pass@1 and 20.6% Pass@10 on *1-shot* MBPP coding benchmarks. Our approach is capable of generating high-quality discrete data in a non-autoregressive fashion, significantly closing the gap between autoregressive models and discrete flow models.

## 1   Introduction

Despite the remarkable success of diffusion and flow models in generating continuous spatial signals such as images (Ho et al., 2020; Rombach et al., 2022; Esser et al., 2024) and videos (Singer et al., 2022; Blattmann et al., 2023), their performance still falters when applied to discrete sequential data compared to autoregressive models. Recent progress in adapting diffusion and flow models to the discrete setting has been made via mostly two approaches: embedding the discrete data in continuous space and applying continuous diffusion (Dieleman et al., 2022; Stark et al., 2024) or designing diffusion or flow processes over discrete state spaces (Austin et al., 2021a; Campbell et al., 2022).

In this paper, we pursue the discrete flow approach of Campbell et al. (2024) and introduce Discrete Flow Matching, a theoretical framework and algorithmic methodology for discrete flow models that yields a state-of-the-art discrete non-autoregressive generative approach. Surprisingly, Discrete Flow Matching exhibits similarities with the continuous Flow Matching (Lipman et al., 2022) approach proposed for continuous signals. Notably, its *generating probability velocity*, employed in the sampling algorithm, is identical in form to its continuous counterpart. Additionally, Discrete Flow Matching offers the following advancements and simplifications over prior methods: It encompasses a more comprehensive family of probability paths transforming source (noise) distributions into target (data) distributions, accommodating arbitrary source-target couplings and time-dependent schedulers. Furthermore, it provides a unified formulation for the generating probability velocity directly expressed in terms of the learned posteriors and schedulers, along with a unified and general theory and algorithm for corrector sampling and iterations. In practice, we observe that path and corrector schedulers are pivotal, and their proper tuning leads to substantial improvements in

```python
def fib(n: int):
    """Return n-th Fibonacci
    number.
    >>> fib(10)
    55
    >>> fib(1)
    1
    >>> fib(8)
    21
    """
    if n < 1: return 0
    if n < 2: return 1

    return fib(n-1) + fib(n-2)
```

```python
def find_position_of_value(arr, x):
    low, mid = 0, 0
    high = len(arr) - 1

    while high >= low:
        mid = (high + low) // 2

        # If x is greater
        if arr[mid] < x:
            low = mid + 1
        # If x is smaller
        elif arr[mid] > x:
            high = mid - 1
        else:
            return mid
    return -1
```

```python
def binary_search(arr, x):
    start = 0
    end = len(arr)-1

    # While performing binary search
    while start <= end:
        mid = (start + end) // 2
        # If x is greater
        if arr[mid] < x:
            start = mid + 1
        # If x is smaller
        elif arr[mid] > x:
            end = mid - 1
        else:
            return mid
    return -1
```

Figure 1: **Code generation examples using Discrete Flow Matching**. Code condition is marked in gray, model generation is marked in yellow. Left sub-figure presents the standard left-to-right prompting; Middle and Right sub-figures, presents complex infilling setup.

generation quality. We have trained a 1.7B parameter Discrete Flow Matching model on the same data mix as in Llama-2 (Touvron et al., 2023) and CodeLlama (Roziere et al., 2023), achieving 6.7% Pass@1 and 13.4% Pass@10 on HumanEval and 6.7% Pass@1 and 20.6% Pass@10 on *1-shot* MBPP coding benchmarks; Figure 1 shows some code generation examples. In conditional text generation our model produces texts with a generative perplexity score of 9.7 as measured by the Llama-3 8B model, surpassing a 1.7B autoregressive model that achieves 22.3 and not far from the Llama-2 7B model that achieves 8.3 in generative perplexity score. We strongly believe that Discrete Flow Matching represents a significant step in bridging the performance gap between discrete diffusion and autoregressive models, and that further enhancements are possible by exploring the vast design space that Discrete Flow Matching has to offer.

## 2 Discrete Flow Matching

### 2.1 Setup and notations

In discrete sequence modeling, we denote a sequence $x$ as an array of $N$ elements $(x^1, x^2, \ldots, x^N)$. Each element, or *token*, within this sequence is selected from a vocabulary of size $d$. Consequently, the entire set of possible sequences is given by $\mathcal{D} = [d]^N$, where $[d] = \{1, \ldots, d\}$. A random variable taking values in the space $\mathcal{D}$ is denoted by $X$ and its corresponding probability mass function (PMF) is $P(X = x)$. For simplicity, throughout the paper, we sometimes omit the random variable $X$ and use $p(x)$ to denote the PMF.

To describe marginalization properties, we denote $p(x^i)$ the $x^i$ marginal of $p$, i.e., $p(x^i) = \sum_{x^{\bar{i}}} p(x)$, where $x^{\bar{i}} = (\ldots, x^{i-1}, x^{i+1}, \ldots) \in [d]^{N-1}$ are all the arguments excluding $i$. Similarly, $p(x^{\bar{i}}) = \sum_{x^i} p(x)$, and $x^i \in [d]$. A useful PMF is the delta function, $\delta_y, y \in \mathcal{D}$, which is defined by

$$\delta_y(x) = \prod_{i=1}^{N} \delta_{y^i}(x^i), \text{ where } \delta_{y^i}(x^i) = \begin{cases} 1 & x^i = y^i \\ 0 & x^i \neq y^i \end{cases}. \tag{1}$$

With the marginal notation $\delta_y(x^i) = \delta_{y^i}(x^i)$ and $\delta_y(x^{\bar{i}}) = \delta_{y^{\bar{i}}}(x^{\bar{i}}) = \prod_{j \neq i} \delta_{y^j}(x^j)$ which simplifies notation.

### 2.2 Source and target distributions

In discrete generative models our goal is to transform source samples $X_0 \sim p$ to target samples $X_1 \sim q$. Our training data, consist of pairs $X_0$ and $X_1$ that are sampled from a joint distribution $\pi(x, y)$, satisfying the marginals constraints $p(x) = \sum_{y \in \mathcal{D}} \pi(x, y), q(y) = \sum_{x \in \mathcal{D}} \pi(x, y)$, i.e.,

$$(X_0, X_1) \sim \pi(X_0, X_1). \tag{2}$$

In the simplest case, the training pairs $X_0$ and $X_1$ are sampled independently from the source and target distributions respectively,

$$(X_0, X_1) \sim p(X_0)q(X_1). \tag{3}$$

***Example:* source and couplings.** Common instantiations of source distribution $p$ are: (i) adding a special token value often referred to as a 'mask' or 'dummy' token, denoted here by $\mathbb{m}$, and setting the source distribution to be all-mask sequences, i.e., $p(x) = \delta_{\mathbb{m}}(x)$; and (ii) using uniform distribution over $\mathcal{D}$, which is equivalent to drawing each $x^i$ independently to be some value in $[d]$ with equal probability, denoted $p(x) = p_{\mathbb{u}}(x)$. In this paper we focus mainly on (i). We further consider two choices of couplings $\pi$: Independent coupling, which we call unconditional coupling (U-coupling), $\pi(x_0, x_1) = p(x_0)q(x_1)$. A random sample that realizes this choice have the form

$$(X_0, X_1) = \Big( (\mathbb{m}, \ldots, \mathbb{m}), X_1 \Big), \tag{4}$$

where $X_1 \sim q(X_1)$ is a random sample from the training set. The second choice of coupling $\pi(x_0, x_1) = p(x_0|x_1)q(x_1)$, which we find improves conditional sampling, partially masks inputs with samples of the form

$$(X_0, X_1) = (\mathbb{l} \odot X_1 + (\mathbf{1} - \mathbb{l}) \odot (\mathbb{m}, \ldots, \mathbb{m}), X_1), \tag{5}$$

where $X_1 \sim q(X_1)$ and $\mathbb{l} \in \{0, 1\}^N$ is a random variable indicating the conditioning, $\odot$ denotes the entry-wise product, and $\mathbf{1} \in \mathbb{R}^N$ is the vector of all ones. We call this conditional coupling (C-coupling).

## 2.3 Probability paths

We follow the Flow Matching approach (Lipman et al., 2022; Liu et al., 2022; Albergo and Vanden-Eijnden, 2022) that uses a predefined *probability path* $p_t$ interpolating $p$ and $q$, i.e.,

$$p_0 = p \quad \text{and} \quad p_1 = q \tag{6}$$

to train the generative model taking a source sample $X_0 \sim p$ to a target sample $X_1 \sim q$. We use arbitrary coupling of source and target (Pooladian et al., 2023; Tong et al., 2023), $\pi(x_0, x_1)$, and the symmetric Flow Matching path (Albergo and Vanden-Eijnden, 2022) to define the marginal probability path,

$$p_t(x) = \sum_{x_0, x_1 \in \mathcal{D}} p_t(x|x_0, x_1)\pi(x_0, x_1), \text{ where } p_t(x|x_0, x_1) = \prod_{i=1}^{N} p_t(x^i|x_0, x_1), \tag{7}$$

and $p_t(x^i|x_0, x_1)$ is a time-dependent probability on the space of tokens $[d]$ conditioned on the pair $x_0, x_1$, and satisfying $p_0(x^i|x_0, x_1) = \delta_{x_0}(x^i)$ and $p_1(x^i|x_0, x_1) = \delta_{x_1}(x^i)$. If the conditional path $p_t(x^i|x_0, x_1)$ satisfies these boundary conditions then the marginal path $p_t(x)$ satisfies equation 6.

In developing the framework, we would like to consider as general as possible set of probability paths that are also tractable to learn within the Flow Matching framework. We consider conditional probability paths as a convex sum of $m$ conditional probabilities $w^j(x^i|x_0, x_1)$, i.e.,

$$p_t(x^i|x_0, x_1) = \sum_{j=1}^{m} \kappa_t^{i,j} w^j(x^i|x_0, x_1), \tag{8}$$

where $\sum_j \kappa_t^{i,j} = 1$ and $\kappa_t^{i,j} \geq 0$ are collectively called the *scheduler*. Note that the scheduler can be defined independently for each location in the sequence $i \in [N]$ or uniformly for all tokens, $\kappa_t^{i,j} = \kappa_t^j$.

A simple yet useful instance of these conditional paths is reminiscent of the continuous Flow Matching paths formulated as convex interpolants,

$$p_t(x^i|x_0, x_1) = (1 - \kappa_t)\delta_{x_0}(x^i) + \kappa_t \delta_{x_1}(x^i), \tag{9}$$

where the scheduler $\kappa_t$ satisfies $\kappa_0 = 0$, $\kappa_1 = 1$, and monotonically increasing in $t$. Another interesting instantiation of equation 8 is adding uniform noise with some probability depending on $t$,

$$p_t(x^i|x_0, x_1) = \kappa_t^1 \delta_{x_1}(x^i) + \kappa_t^2 p_{\mathbb{u}}(x^i) + \kappa_t^3 \delta_{x_0}(x^i), \tag{10}$$

where $\kappa_0^1 = 0$, $\kappa_1^1 = 1$, $\kappa_0^2 = \kappa_1^2 = 0$ (remembering that $\sum_j \kappa_t^{i,j} = 1$ and $\kappa_t^{i,j} \geq 0$).

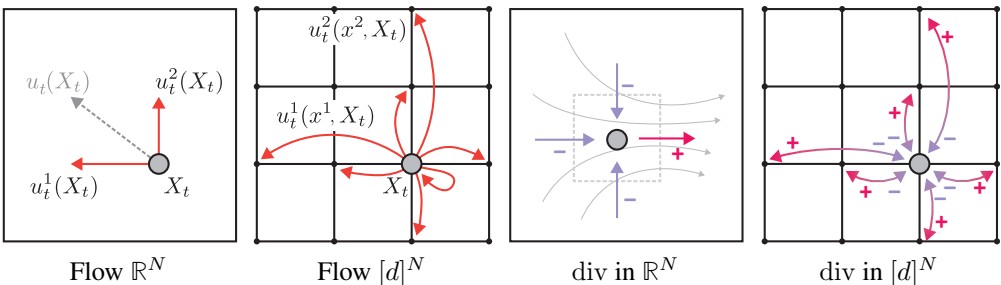

Figure 2: Discrete flow in $\mathcal{D} = [d]^N$ with $d = 4, N = 2$ (middle-left) versus continuous flow in $\mathbb{R}^N$, $N = 2$ (left). The rate of change of probability of a state (gray disk) is given by the divergence operator shown in the continuous case (middle right) and the discrete case (right).

## 2.4 Generating Probability Velocities

**Continuous generating velocity.** Sampling in continuous Flow Matching is performed by updating the current (continuous) sample $X_t \in \mathbb{R}^N$, $t \in [0, 1)$, according to a learned *generating velocity field* $u_t^i(X_t)$, $i \in [N]$. Euler sampling follows the (deterministic) rule

$$X_{t+h}^i = X_t^i + h u_t^i(X_t), \tag{11}$$

where $h > 0$ is a user-defined time step. Note that equation 11 is updating separately each of the sample coordinates, $X_t^i$, $i \in [N]$, see e.g., Figure 2, left. The velocity $u_t^i(X_t)$ can be either directly modeled with a neural network, or parameterized via the *denoiser* (a.k.a. $x$-prediction) or *noise-prediction* (a.k.a. $\varepsilon$-prediction), see left column in Table 1. If, for all $t \in [0, 1)$, starting at $X_t \sim p_t$ and sampling with equation 11 provides $X_{t+h} \sim p_{t+h} + o(h)$[1] then we say that $u_t$ *generates* $p_t$.

**Generating probability velocity.** For defining Flow Matching in the discrete setting, we follow Campbell et al. (2024) and consider a Continuous-Time discrete Markov Chain (CTMC) paradigm, namely the sample $X_t$ is jumping between states in $\mathcal{D}$, depending on a continuous time value $t \in [0, 1]$. Similar to the continuous Flow Matching setting described above, we focus on a model that predicts the rate of probability change of the current sample $X_t$ in each of its $N$ tokens, see Figure 2, middle-left. Then, each token of the sample $X_t \sim p_t$ is updated independently by

$$X_{t+h}^i \sim \delta_{X_t^i}(\cdot) + h u_t^i(\cdot, X_t), \tag{12}$$

where we call $u_t$ the *probability velocity* as reminiscent of the velocity field in continuous Flow Matching, and as in the continuous case, we define:

**Definition 1.** Probability velocity $u_t$ *generates* the probability path $p_t$ if, for all $t \in [0, 1)$ and given a sample $X_t \sim p_t$, the sample $X_{t+h}$ defined in equation 12 satisfies $X_{t+h} \sim p_{t+h} + o(h)$.

Algorithm 1 formulates a basic sampling algorithm given a generating probability velocity $u_t$. In order for the r.h.s. of equation 12 to define a proper PMF for sufficiently small $h > 0$, it is necessary and sufficient that the probability velocity satisfies the conditions

$$\sum_{x^i \in [d]} u_t^i(x^i, z) = 0, \text{ and } u_t^i(x^i, z) \geq 0 \text{ for all } i \in [N] \text{ and } x^i \neq z^i. \tag{13}$$

Now the main question is how to find a probability velocity $u_t$ that generates the probability path defined in equations 7 and 8? A key insight in Flow Matching (Lipman et al., 2022) is that $u_t$ can be constructed as a marginalization of *conditional* probability velocities, $u_t^i(x^i, z|x_0, x_1)$,

---

**Algorithm 1** Flow Matching sampling.
---
**Require:** velocity $u_t$, sample $X \sim p$, step size $h = \frac{1}{n}$
  **for** $t = 0, h, 2h, \ldots, 1 - h$ **do**
    $X^i \sim \delta_{X^i}(\cdot) + h u_t^i(\cdot, X)$, for $i \in [N]$   ▷ eq. 24 or 22
  **end for**
  **return** $X$
---

generating the corresponding conditional probability paths $p_t(x^i|x_0, x_1)$. This can also be shown to hold in the discrete CTMC setting (Campbell et al., 2024), where a reformulation in our context and notation is as follows.

---

[1]The $o(h^\ell)$ notation means a function going to zero faster than $h^\ell$ as $h \to 0$, i.e., $\frac{o(h^\ell)}{h^\ell} \xrightarrow{h \to 0} 0$.

**Theorem 2.** *Given a conditional probability velocity $u_t^i(x^i, z|x_0, x_1)$ generating a conditional probability path $p_t(x|x_0, x_1)$, the marginal velocity defined by*

$$u_t^i(x^i, z) = \sum_{x_0, x_1 \in \mathcal{D}} u_t^i(x^i, z|x_0, x_1) p_t(x_0, x_1|z) \tag{14}$$

*generates the marginal probability path $p_t(x)$, where by Bayes' rule*

$$p_t(x_0, x_1|z) = \frac{p_t(z|x_0, x_1)\pi(x_0, x_1)}{p_t(z)}. \tag{15}$$

For completeness we provide a simple proof of this theorem in Appendix E.2. The proof, similar to the continuous Flow Matching case, shows that $u_t$ and $p_t$ satisfy the (discrete version of the) Continuity Equation.

**The Continuity Equation.** To provide the mathematical tool for showing that a probability velocity $u_t$ does indeed generate the probability path $p_t$, and also to further highlight the similarities to the continuous case, we next formulate the *Kolmogorov Equations*, which describe the state probability rate $\dot{p}_t(x)$, $x \in \mathcal{D}$, in CTMC as a Continuity Equation (CE). The Continuity Equation, similarly to Kolmogorov Equations, describes $\dot{p}_t(x)$, $x \in \mathbb{R}^N$ in the *continuous case*, and is formulated as the Partial Differential Equation (PDE)

$$\dot{p}_t(x) + \mathrm{div}_x(p_t u_t) = 0, \tag{16}$$

where the divergence operator $\mathrm{div}_x(v)$ applied to a vector field $v : \mathbb{R}^N \to \mathbb{R}^N$ is defined by

$$\mathrm{div}_x(v) = \sum_{i=1}^N \partial_{x^i} v^i(x), \tag{17}$$

and intuitively means the total flux leaving $x$, see Figure 2 (middle-right). This gives an intuitive explanation to the Continuity Equation: the rate of the probability $\dot{p}_t(x)$ of a state $x \in \mathbb{R}^N$ equals the total *incoming probability flux*, $p_t u_t$, at $x$. In the discrete case (CTMC) the Continuity Equation (equation 16) holds as is, once the discrete divergence operator is properly defined, i.e., to measure the outgoing flux from a discrete state. In more detail, given some vector field, which in the discrete case is a scalar-valued function over pairs of states, $v : \mathcal{D} \times \mathcal{D} \to \mathbb{R}$, the discrete divergence is

$$\mathrm{div}_x(v) = \sum_{z \in \mathcal{D}} [v(z, x) - v(x, z)], \tag{18}$$

where $v(z, x)$ represents the flux $x \to z$ and $v(x, z)$ represent the opposite flux $z \to x$; see Figure 2, right. Now, in our case (see Figure 2, middle-left), the probability flux at a state $x \in \mathcal{D}$ involves all sequences with at most one token difference from $x$, i.e., the probability flux $p_t u_t$ at $x$ takes the form $v(x, z) = p_t(z)u_t^i(x^i, z)$ and $v(z, x) = p_t(x)u_t^i(z^i, x)$ for $z$ and $x$ that differ only in the $i$-th token, $v(x, x) = p_t(x)\sum_{i=1}^N u_t^i(x^i, x)$, and $v(x, z) = 0$ for all other $(z, x) \in \mathcal{D} \times \mathcal{D}$. A direct calculation now shows (see Appendix E.1):

$$\mathrm{div}_x(p_t u_t) = -\sum_{z \in \mathcal{D}} p_t(z) \left[ \sum_{i=1}^N \delta_z(x^{\bar{i}}) u_t^i(x^i, z) \right]. \tag{19}$$

Checking that a probability velocity $u_t$ generates a probability path $p_t$ (in the sense of Definition 1) amounts to verifying the Continuity Equation (equation 16). Indeed, using arguments from Campbell et al. (2024) and the discrete divergence operator, the PMF of $X_{t+h}$ defined by sampling according to equation 12 is

$$\mathbb{E}_{X_t} \prod_{i=1}^N \left[ \delta_{X_t}(x^i) + h u_t^i(x^i, X_t) \right] = \mathbb{E}_{X_t} \left[ \delta_{X_t}(x) + h \sum_{i=1}^N \delta_{X_t}(x^{\bar{i}}) u_t^i(x^i, X_t) \right] + o(h) \tag{20}$$

$$= p_t(x) - h\,\mathrm{div}_x(p_t u_t) + o(h) \overset{(16)}{=} p_t(x) + h\dot{p}_t(x) + o(h) = p_{t+h}(x) + o(h),$$

where we assume $X_t \sim p_t$, the first equality uses the identity $\prod_i [a^i + hb^i] = \prod_i a^i + h \sum_i (\prod_{j \neq i} a^j) b^i + o(h)$, the second equality uses equation 19, and the previous-to-last equality uses the Continuity Equation (equation 16). This shows that if the Continuity Equation holds then $u_t$ generates $p_t$ in the sense of Definition 1.

Table 1: Generating (marginal) velocity fields have identical form for the continuous and discrete Flow Matching when using denoiser/noise-prediction parameterization; $\hat{x}_{1|t}(z) = \mathbb{E}_{X_1 \sim p_t(\cdot|z)} X_1$ is the standard continuous denoiser (a.k.a. $x$-prediction) and $\hat{x}_{0|t}(z) = \mathbb{E}_{X_0 \sim p_t(\cdot|z)} X_0$ is the standard noise-prediction (a.k.a. $\epsilon$-prediction).

| | **Continuous** Flow Matching | **Discrete** Flow Matching |
|---|---|---|
| Marginal prob. | $p_t(x) = \sum_{x_0,x_1} \prod_{i=1}^{N} p_t(x^i|x_0,x_1)\pi(x_0,x_1)$ | |
| Conditional prob. | $p_t(x^i|x_0,x_1) = \delta_{\kappa_t x_1 + (1-\kappa_t)x_0}(x^i)$ | $p_t(x^i|x_0,x_1) = \kappa_t \delta_{x_1}(x^i) + (1-\kappa_t)\delta_{x_0}(x^i)$ |
| VF-*Denoiser* | $u_t^i(X_t) = \frac{\dot{\kappa}_t}{1-\kappa_t}\left[\hat{x}_{1|t}^i(X_t) - X_t^i\right]$ | $u_t^i(x^i, X_t) = \frac{\dot{\kappa}_t}{1-\kappa_t}\left[p_{1|t}(x^i|X_t) - \delta_{X_t}(x^i)\right]$ |
| VF-*Noise-pred* | $u_t^i(X_t) = \frac{\dot{\kappa}_t}{\kappa_t}\left[X_t^i - \hat{x}_{0|t}^i(X_t)\right]$ | $u_t^i(x^i, X_t) = \frac{\dot{\kappa}_t}{\kappa_t}\left[\delta_{X_t}(x^i) - p_{0|t}(x^i|X_t)\right]$ |

**Conditional and marginal generating velocities.** We provide the probability velocities generating the conditional probability paths $p_t(x|x_0, x_1)$ defined in equations 7 and 8. Then, using the marginalization formula in equation 14 we end up with a closed-form marginal velocity for the probability paths $p_t(x)$. In Appendix E.3 we show

**Theorem 3** (Probability velocity of conditional paths). *A generating probability velocity for the conditional paths $p_t(x|x_0, x_1)$ defined in equations 7 and 8 is*

$$u_t^i(x^i, z|x_0, x_1) = \sum_{j=1}^{m} a_t^{i,j} w^j(x^i|x_0, x_1) + b_t^i \delta_z(x^i), \tag{21}$$

*with $a_t^{i,j} = \dot{\kappa}_t^{i,j} - \kappa_t^{i,j} \dot{\kappa}_t^{i,\ell}/\kappa_t^{i,\ell}$, and $b_t^i = \dot{\kappa}_t^{i,\ell}/\kappa_t^{i,\ell}$ where $\ell = \arg\min_{j \in [m]}\left[\dot{\kappa}_t^{i,j}/\kappa_t^{i,j}\right]$.*

Now, computing the marginal probability velocity using equation 14 applied to the conditional probability velocity in equation 21 gives

$$u_t^i(x^i, z) = \sum_{j=1}^{m} a_t^{i,j} \hat{w}_t^j(x^i, z) + b_t^{i,j} \delta_z(x^i), \tag{22}$$

where the posteriors $\hat{w}_t^j$ of $w^j$ (that are later shown to be tractable to learn) are defined by

$$\hat{w}_t^j(x^i, z) = \sum_{x_0, x_1 \in \mathcal{D}} w^j(x^i|x_0, x_1) p_t(x_0, x_1|z), \tag{23}$$

where $p_t(x_0, x_1|z)$ (defined in equation 15) is the posterior probability of $x_0, x_1$ conditioned on the current state $X_t = z$. A useful instantiation of the general velocity in equation 22 is when considering the path family in equation 9, for which $w^1(x^i|x_0, x_1) = \delta_{x_1}(x^i)$, $w^2(x^i|x_0, x_1) = \delta_{x_0}(x^i)$, $\kappa_t^{i,1} = \kappa_t$, $\kappa_t^{i,2} = 1 - \kappa_t$, $\dot{\kappa}_t \geq 0$ (i.e., monotonically non-decreasing in $t$) and in this case equation 22 reads as

$$u_t^i(x^i, z) = \frac{\dot{\kappa}_t}{1 - \kappa_t}\left[p_{1|t}(x^i|z) - \delta_z(x^i)\right] \tag{24}$$

where we use the notation $p_{1|t}(x^i|z) = \sum_{x_0,x_1} \delta_{x_1}(x^i) p_t(x_0, x_1|z)$ for the *probability denoiser*.

**Sampling backward in time.** We can also sample *backwards in time* by following the sampling rule $X_{t-h}^i \sim \delta_{X_t^i}(\cdot) - h u_t^i(\cdot, X_t)$. In this case $-u_t^i(x^i, z)$ should satisfy equation 13. A (backward-time) generating probability velocity can then be achieved from equation 22 with the simple change to the coefficients $a_t^{i,j}$ and $b_t^{i,j}$, see Appendix E.4. For $p_t$ defined with equation 9 the generating velocity is

$$u_t^i(x^i, z) = \frac{\dot{\kappa}_t}{\kappa_t}\left[\delta_z(x^i) - p_{0|t}(x^i|z)\right], \tag{25}$$

where in this case $p_{0|t}(x^i|z) = \sum_{x_0,x_1 \in \mathcal{D}} \delta_{x_0}(x^i) p_t(x_0, x_1|z)$ is the *probability noise-prediction*.

Remarkably, the generating velocity fields in 24 and 25 take the *exact same form* as the generating (a.k.a. marginal) velocity fields in *continuous* flow matching when parameterized via the denoiser or noise-prediction parameterizations and using the same schedulers, see Table 1 and Appendix E.9 for explanation of the continuous case. In Appendix E.4 we provide the backward-time version of Theorem 3.

**Corrector sampling.** Combining the forward-time $\hat{u}_t$ ( equation 24) and backward-time $\check{u}_t$ ( equation 25), i.e.,

$$\bar{u}_t^i(x^i, z) = \alpha_t \hat{u}_t^i(x^i, z) - \beta_t \check{u}_t^i(x^i, z), \tag{26}$$

provides a valid forward-time probability velocity field (i.e., satisfies equation 13) for $t \in (0,1)$ as long as $\alpha_t, \beta_t > 0$. This velocity field can be used for two types of corrector sampling: (i) When $\alpha_t - \beta_t = 1$ sampling with $\bar{u}_t$ leads to *corrector sampling* where intuitively each step moves $1 + \beta_t$ forward in time and $-\beta_t$ backwards, which allows reintroducing noise into the sampling process; and (ii) when $\alpha_t - \beta_t = 0$ sampling with $\bar{u}_t$ when fixing $t \in (0,1)$ leads to *corrector iterations* where limit samples distribute according to $p_t$. In Appendix E.6 we prove:

**Theorem 4.** *For perfectly trained posteriors and $\alpha_t, \beta_t > 0$, $t \in (0,1)$, $\bar{u}_t$ in equation 26 is a probability velocity, i.e., satisfies equation 13, and: (i) For $\alpha_t - \beta_t = 1$, $\bar{u}_t$ provides a probability velocity generating $p_t$; (ii) For $\alpha_t - \beta_t = 0$, repeatedly sampling with $\bar{u}_t$ at fixed $t \in (0,1)$ and sufficiently small $h$ is guaranteed to converge to a sample from $p_t$.*

One simplification to equation 26 can be done in the case of paths constructed with conditional as in equation 9, independent coupling $\pi(x_0, x_1) = p(x_0)q(x_1)$, and i.i.d. source $p(x_0) = \prod_{i=1}^N p(x_0^i)$, e.g., $p(x_0^i)$ is uniform over $[d]$ or $\delta_{\mathfrak{m}}(x_0^i)$. In this case, the backward-time formula in equation 25 take an equivalent simpler form

$$\check{u}_t^i(x^i, z) = \frac{\dot{\kappa}_t}{\kappa_t} \left[ \delta_z(x^i) - p(x^i) \right], \tag{27}$$

which does not require estimation of the posterior $p_{0|t}$. See Appendix E.5 for the derivation.

**Training.** Equation 22 shows that for generating samples from a probabilty path $p_t(x)$ we require the posteriors $\hat{w}_t^j(x^i|X_t)$. Training such posteriors can be done by minimizing the loss

$$\mathcal{L}(\theta) = - \sum_{j \in [m], i \in [N]} \mathbb{E}_{t, (X_0, X_1), X_t, Y_j^i} \log \hat{w}_t^j(Y_j^i|X_t; \theta), \tag{28}$$

where $t$ is sampled according to some distribution in $[0,1]$ (we used uniform), $(X_0, X_1) \sim \pi(X_0, X_1)$, $X_t \sim p_t(X_t|X_0, X_1)$, and $Y_j^i \sim w^j(Y_j^i|X_0, X_1)$; $\theta \in \mathbb{R}^p$ denotes the learnable parameters. In the common case we use in this paper of learning a single posterior, i.e., the probability denoiser $p_{1|t}$, the loss takes the form $\mathcal{L}(\theta) = - \sum_{i \in [N]} \mathbb{E}_{t, (X_0, X_1), X_t} \log p_{1|t}(X_1^i|X_t)$. In Appendix E.7 we prove:

**Proposition 5.** *The minimizer of $\mathcal{L}$ (equation 28) is $\hat{w}_t^j(x^i|X_t)$ (equation 23).*

## 3 Related work

In the section we cover the most related work to ours; in Appendix A we cover other related work.

**Discrete Flows (Campbell et al., 2024)** is probably the most related work to ours. We build upon their CTMC framework and offer the following generalizations and simplifications: We consider arbitrary couplings $(X_0, X_1)$, and offer a novel and rather general family of probability paths (equation 8) for which we provide the generating probability velocities in a unified closed-form formula (equations 22-25). These in particular recreate the same formulas as the continuous Flow Matching counterpart (Table 1). We furthermore develop a general corrector velocity (equation 26) that unifies both corrector iterations (Song et al., 2020; Campbell et al., 2022) and stochastic sampling of Campbell et al. (2024). We show that particular choices of noise schedulers $\kappa_t$ ($\kappa_t = t$ reproduces Campbell et al. (2024)) and corrector schedulers provide a boost in results. Lastly, we opted for the term *probability velocity* for $u_t^i(x^i, X_t)$ as it is not precisely a rate matrix in the state space $\mathcal{D} \times \mathcal{D}$ used in CTMC since $u_t^i(x^i, z)$ for all $i \in [N]$ define multiple self-edges $z \to z$.

**Masked modeling (Ghazvininejad et al., 2019; Chang et al., 2022).** In case of a masked model, i.e., when the source distribution is $p(x) = \delta_{\mathfrak{m}}(x)$, we achieve an interesting connection with MaskGit

Table 2: Generative perplexity on unconditional text generation compared to prior work. All models are sampled without the use of temperature or corrector steps. Double precision sampling results are reported in Table 5.

| METHOD | NFE | LLAMA-2↓ | LLAMA-3↓ | GPT2↓ | ENTROPY |
|---|---|---|---|---|---|
| Data | - | 7.0 | 9.4 | 14.7 | 7.7 |
| Autoregressive | 1024 | 31.4 | 54.8 | 45.3 | 7.1 |
| Savinov et al. (2021) | 200 | 29.5 | 45.1 | 34.7 | 5.2 |
| Austin et al. (2021a) | 1000 | 697.6 | 768.8 | 837.8 | 7.6 |
| Han et al. (2022) | >10000 | 73.3 | 203.1 | 99.2 | 4.8 |
| Lou et al. (2023) | 256/512/1024 | 38.6/33.7/27.2 | 69.2/58.6/43.9 | 64.3/53.4/40.5 | 7.8/7.7/7.6 |
| Campbell et al. (2024) | 256/512/1024 | 38.5/33.5/28.7 | 69.0/56.5/46.5 | 65.2/53.3/43.0 | 7.8/7.7/7.6 |
| **FM (equation 9)** | 256/512/1024 | 34.2/30.0/22.5 | 58.5/48.8/33.8 | 54.2/43.5/29.3 | 7.7/7.6/7.2 |
| **FM (equation 10)** | 256/512/1024 | 30.0/27.5/22.3 | 48.2/43.5/31.9 | 47.7/41.8/28.1 | 7.6/7.5/7.1 |

Table 3: Generative perplexity on conditional text generation.

| METHOD | MODEL SIZE | NFE | LLAMA-2↓ | LLAMA-3↓ | ENTROPY |
|---|---|---|---|---|---|
| Llama-3 (Reference) | 8B | 512 | 6.4 | 7.3 | 6.8 |
| Llama-2 (Reference) | 7B | 512 | 5.3 | 8.3 | 7.1 |
| Autoregressive | 1.7B | 512 | 14.3 | 22.3 | 7.2 |
| Savinov et al. (2021) | 1.7B | 200 | 10.8 | 15.4 | 4.7 |
| **FM (U-coupling)** | 1.7B | 256/512 | 10.7/9.5 | 11.2/10.3 | 6.7/6.7 |
| **FM (C-coupling)** | 1.7B | 256/512 | 10.2/8.9 | 10.0/9.7 | 6.8/6.7 |

showing it is actually an instance of Discrete Flow Matching with a small yet crucial change to its sampling algorithm. First, in Appendix E.8 we prove that in the masked setting, the probability denoiser $p_{1|t}$ is *time-independent*:

**Proposition 6.** *For paths defined by equations 7 and 9 with source $p(x) = \delta_{\mathtt{m}}(x)$ the posterior $p_t(x_0, x_1|z) = p(x_0, x_1|z)$ is time-independent. Consequently, the probability denoiser $p_{1|t}(x^i|z) = p_1(x^i|z)$ is also time-independent.*

This shows that the probability denoiser can be learned with no time dependence, similar to the unmasking probabilities in MaskGit. During sampling however, there are two main differences between our sampling and MaskGit sampling. First, unmasking of tokens in our algorithm is done according to the probability $\delta_{X_t}(x^i) + h u_t^i(x^i, X_t)$ *independently* for each token $x^i$, $i \in [N]$. This procedure is justified as it samples from the correct probability asymptotically via the derivation of the Continuity Equation 20. This is in contrast to MaskGit that prioritizes the token to be unmasked according to some *confidence*. In the experiments section we show that MaskGit's prioritization, although has some benefit in the very low NFE regime, is actually introducing a strong bias in the sampling procedure and leads to inferior overall results. Secondly, using corrector sampling allows for reintroducing masks to already unmasked tokens in a way that is still guaranteed to produce samples from $p_t$, see Theorem 4; we find this to have a significant positive effect on the generation quality.

**Discrete diffusion.** D3PM (Austin et al., 2021a) and Argmax flows (Hoogeboom et al., 2021) introduced diffusion in discrete spaces by proposing a corruption process for categorical data. A later work by Campbell et al. (2022) introduced discrete diffusion models with continuous time, and Lou et al. (2023) proposed learning probability ratios, extending score matching (Song and Ermon, 2019) to discrete spaces.

## 4 Experiments

We evaluate our method on the tasks of language modeling, code generation, and image generation. For language modeling, we compare the proposed method against prior work considering the widely used generative perplexity metric. We scale the models to 1.7 billion parameters and present results on coding tasks, i.e., HumanEval (Chen et al., 2021), MBPP (Austin et al., 2021b), demonstrating

Table 4: Execution based code generation evaluation.

| METHOD | DATA | HUMANEVAL↑ | | | MBPP (1-SHOT)↑ | | |
|---|---|---|---|---|---|---|---|
| | | Pass@1 | Pass@10 | Pass@25 | Pass@1 | Pass@10 | Pass@25 |
| Autoregressive | Text | 1.2 | 3.1 | 4.8 | 0.2 | 1.7 | 3.3 |
| | Code | 14.3 | 21.3 | 27.8 | 17.0 | 34.3 | 44.1 |
| **FM** | Text | 1.2 | 2.6 | 4.0 | 0.4 | 1.1 | 3.6 |
| | Code | 6.7 | 13.4 | 18.0 | 6.7 | 20.6 | 26.5 |
| **FM (Oracle length)** | Code | 11.6 | 18.3 | 20.6 | 13.1 | 28.4 | 34.2 |

the most promising results to date in a non-autoregressive context. In image generation, we present results for a fully discrete CIFAR10 (Krizhevsky et al., 2009). Further details of the experimental setup for each model are provided in Appendix G.

**Experimental setup.** In our experiments we used the masked source, i.e., $p = \delta_\mathfrak{m}$, and trained with both unconditional coupling (U-coupling, equation 4) and conditional couplings (C-coupling, equation 5) with the probability path defined in equations 7, 9 and in one case 10. We trained a probability denoiser (loss in equation 28) and sampled using the generating velocity in equation 24 and Algorithm 1. We used a particular choice of probability path scheduler $\kappa_t$, as well as corrector steps defined by a scheduler $\alpha_t$ and temperature annealing. We found the choice of these schedulers to be pivotal for the model's performance. In Appendix D we perform an ablation study, evaluating various scheduler choices.

## 4.1 Language modeling

We experimented with our method in three settings: (i) Small model (150M parameters) - comparison to other non-autoregressive baselines in unconditional text generation; (ii) Large model (1.7B parameters) - comparison to autoregressive models in conditional text generation; and (iii) Large model (1.7B parameters) - conditional code generation. As computing exact likelihood for non-autoregressive model is a challenge, for (i),(ii) we use the generative perplexity metric (Appendix G measured with GPT2 (Radford et al., 2019), Llama-2 (Touvron et al., 2023), and Llama-3, and we also monitor the sentence entropy (Appendix G) to measure diversity of tokens and flag repetitive sequences, which typically yield low perplexity. Throughout our experiments we noticed entropy $\geq 6$ usually corresponds to diverse texts. For (iii) we evaluated using the success rate of coding tasks.

**Evaluation against prior work.** We evaluate our method against prior work on non-autoregressive modeling. For a fair comparison, all methods are trained on a 150M parameters models using the OpenWebText (Gokaslan and Cohen, 2019) dataset. We also fix all sampling hyperparameters to the most basic settings, i.e., no temperature, top probability, corrector steps, etc. For our method we tried two paths defined by equations 9 and 10. Results are reported in Table 2, where our method outperforms all baselines in generative perplexity for all numbers of function evaluations (NFE).

**Conditional text generation.** In this experiment, we train both C-coupling and U-coupling 1.7B parameters FM models with paths defined by equation 9 on a large scale data mix (Touvron et al., 2023). Table 3 presents the generative perplexity of conditional generations from our method; the conditions we used are the prefixes of the first 1000 samples in OpenWeb dataset. We also compare to existing state-of-the-art autoregressive models. Our results demonstrate that our model effectively narrows the gap in generative perplexity with autoregressive models, while maintaining an entropy comparable to the recent Llama-3 8B model. Furthermore, we note the C-coupling trained model produces slightly better perplexity in conditional tasks than the U-coupling model. In Appendix I we present qualitative conditional samples produced by our U-coupling model.

**Code generation.** Here we trained our basic setting of a 1.7B parameters FM model with U-coupling and path as in equation 9 on a code-focused data mix (Roziere et al., 2023). Table 4 presents results on HumanEval and MBPP (1-shot) for pass@$\{1, 10, 25\}$. In Table 4, 'Oracle length' evaluates the performance of our model when conditioning on the length of the solution. This is done by inserting an 'end of text' token in the same position of the ground truth solution. Our method achieves non-trivial results on both tasks, which to the best of our knowledge is the first instance of a non-autoregressive method being capable of non-trivial coding tasks. In Appendix C, we analyze the

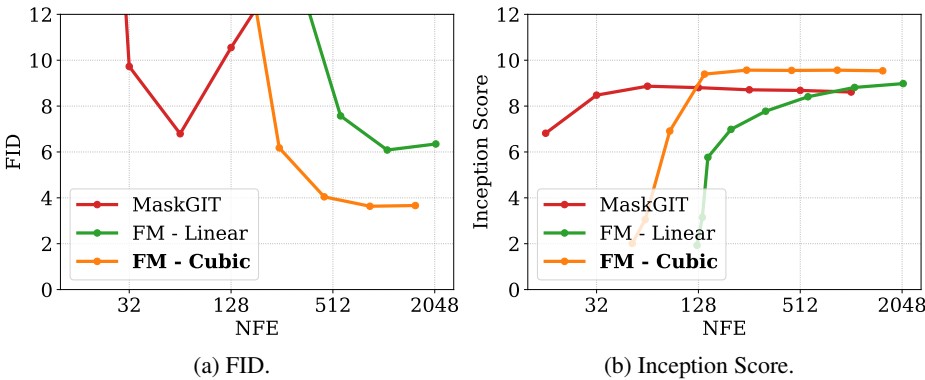

(a) FID.

(b) Inception Score.

Figure 3: FID and Inception scores vs. number of function evaluations (NFE).

proposed method for code infilling, which can be achieved as our model allows non-autoregressive generation. Lastly, in Appendix H we show qualitative examples of success and failure cases produced by our model on the coding tasks, and in Appendix H.3 we show examples of code infilling.

## 4.2 Image generation

We performed a fully discrete image generation, without using any metric or neighboring information between color values. We trained an FM model with U-coupling and path as in equation 9 on CIFAR10 to predict discrete color value for tokens, i.e., $d = 256$, with sequence length of $N = 32 \times 32 \times 3$. For generative quality we evaluate the Fréchet Inception Distance (FID) (Heusel et al., 2017). Ablations for the probability path schedulers are provided in Figure 8 in the Appendix G. In Figure 3a we compare our method with: (i) MaskGIT (Chang et al., 2022); and (ii) (Campbell et al., 2024) which coincides with our method for a linear scheduler. More details in Appendix G. As can be seen in the Figure 3a, our method outperforms both baselines, achieving 3.63 FID at $1024$ NFE. In fig. 3b we observe a similar trend when evaluating Inception score. As discussed above, MaskGit sampling performs better for low NFE but quickly deteriorates for higher NFE. We attribute this to a bias introduced in the sampling process via the confidence mechanism.

## 5 Conclusions and future work

We introduce Discrete Flow Matching, a generalization of continuous flow matching and discrete flows that provides a large design space of discrete non-autoregressive generative models. Searching within this space we were able to train large scale language models that produce generated text with an improved generative perplexity compared to current non-autoregressive methods and able to solve coding tasks at rates not achievable before with non-autoregressive models, as far as we are aware. While reducing the number of network evaluations required to generate a discrete sample compared to autoregressive models, Discrete Flow Matching still does not achieve the level of sampling efficiency achieved by its continuous counterpart, flagging an interesting future work direction. Another interesting direction is to explore the space of probability paths in equation 8 (or a generalization of which) beyond what we have done in this paper. We believe discrete non-autoregressive models have the potential to close the gap and even surpass autoregressive models as well as unlock novel applications and use cases. As our work introduces an alternative modeling paradigm to discrete sequential data such as language and code, we feel it does not introduce significant societal risks beyond those that already exist with previous large language models.

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

# A Related works, continuation

We provide here some more details on relevant related works.

**Continuous diffusion and flows.** Another line of works has been exploring the use of continuous space diffusion for discrete data, typically operating in the logits space (Dieleman et al., 2022; Li et al., 2022; Han et al., 2022; Lin et al., 2022; Chen et al., 2022). An additional body of work has been focusing on the adoption of latent diffusion-like modeling (Lovelace et al., 2022; He et al., 2022). Stark et al. (2024) proposed to learn a continuous Flow Matching on the probability simplex with Dirichlet paths.

**Autoregressive modeling.** Autoregressive models have been a significant area of focus in recent years, particularly in the context of natural language processing and machine learning (Zhao et al., 2023). Autoregressive modeling, in its most fundamental form, utilizes the chain rule to learn the joint sequence probability by breaking it down into next-token conditional probabilities. GPT-2 (Radford et al., 2019), showcased the power of autoregressive language models in generating coherent and contextually relevant text over long passages. Its successor, GPT-3 (Brown et al., 2020), further pushed the boundaries, demonstrating impressive performance across a range of tasks without task-specific training data. Later models were adapted to other domains such as, code (Roziere et al., 2023; Li et al., 2023; Chen et al., 2021), biology (Zhang et al., 2024; Ferruz and Höcker, 2022; Madani et al., 2023), math (Romera-Paredes et al., 2024; Imani et al., 2023; Ahn et al., 2024), audio (Kreuk et al., 2022; Copet et al., 2024; Hassid et al., 2024) and more.

**Masked generative modeling.** Masked generative modeling proposes to mask a variable portion of the input sequence and training a model to predict this masked section. Ghazvininejad et al. (2019) proposed Mask-Predict, a masked language modeling with parallel decoding. Savinov et al. (2021) extended the mask-modeling approach by employing an additional loss term that incorporates rolling model predictions. MaskGIT (Chang et al., 2022) followed a similar path, for the task of class-conditioned image synthesis, Chang et al. (2023) extended this approach to high-quality textually guided image generation over low-resolution images followed by a super-resolution module. Recently, Ziv et al. (2024) proposed a text-to-music method, which relies on the MaskGIT foundations while observing that span masking boosts the quality of the generated sequence significantly.

# B Further implementation details

**Safe sampling.** When sampling according to Algorithm 1 using the generating probability velocity in equation 22, an arbitrary step size $h > 0$ can make some probabilities in $\delta_{X_t^i}(\cdot) + h u_t^i(\cdot, X_t)$ negative and consequently require clamping and injecting further error into the sampling process that can in turn accumulate to a non-negligible global sampling error. A simple fix that guarantees a valid probability distribution while keeping the $o(h)$ sampling error at the relatively manageable price of potentially more function evaluations is using the following adaptive step size in Algorithm 1: at time $t \in [0, 1)$ use

$$h_{\text{adaptive}} = \min \left\{ h, \min_i \left| \frac{\kappa_t^{i,\ell}}{\dot{\kappa}_t^{i,\ell}} \right| \right\}. \tag{29}$$

As can be verified with the general probability velocity formula in equation 22, the above choice for $h_{\text{adaptive}}$ guarantees $\delta_{X_t^i}(\cdot) + h u_t^i(\cdot, X_t)$ is a valid PMF. As mostly used in this paper, for the probability denoiser parameterization (equation 24) the adaptive step is

$$h_{\text{adaptive}} = \min \left\{ h, \frac{1 - \kappa_t}{\dot{\kappa}_t} \right\}. \tag{30}$$

With the corrector sampling (equations 26 and 51) we have the adaptive step:

$$h_{\text{adaptive}} = \min \left\{ h, \left[ \frac{\alpha_t \dot{\kappa}_t}{1 - \kappa_t} + \frac{\beta_t \dot{\kappa}_t}{\kappa_t} \right]^{-1} \right\}. \tag{31}$$

**Conditioning.**  In our unconditional coupling (U-coupling), see equation 5, we define the conditioning pattern based on prefixes of random length $N_0 < N$, i.e.,

$$\mathbb{1} = (\overbrace{1,\ldots,1}^{N_0}, \overbrace{0,\ldots,0}^{N-N_0}).$$

During the training phase, we sample $N_0 \sim \mathcal{U}(0, N)$ and adjust the input sequence in accordance with the mask $\mathbb{1}$.

During conditional sampling with Algorithm 1 we replace, after each update step, the relevant tokens with the conditioned ones, i.e., $\tilde{X} = \mathbb{1} \odot Y + (\mathbf{1} - \mathbb{1}) \odot X$, where $X$ is the current sample, $Y$ is the condition, and $\mathbb{1}$ is the condition's mask.

**NFE bound.**  For mask modeling, i.e., $p = \delta_{\mathrm{m}}$, we have seen that the probability denoiser is time-independent (see Proposition 6). Consequently, when sampling with Algorithm 1 and $u_t$ from equation 24 without corrector sampling one is not required to recompute the forward pass $p_{1|t}(\cdot|X_t)$ if $X_t$ is identical to $X_{t-h}$ (i.e., no $\mathrm{m}$ has been unmasked). This means that the NFE of Algorithm 1 in this case is bounded by the number of tokens $N$.

**Post training scheduler change.**  For a trained posterior $\hat{w}_t(x^i|z)$ of a conditional probability path as in equation 9 with a scheduler $\kappa_t$, the velocity is given by equations 24 or 25, where $\hat{w}_t(x^i|z)$ is either $p_{1|t}(x^i|z)$ or $p_{0|t}(x^i|z)$ respectively. In this case, we can apply the velocities in equations 24 and 25 for sampling with any scheduler $\kappa'_t$, using the change of scheduler formula for posteriors,

$$\hat{w}'_t(x^i|z) = \hat{w}_{t'}(x^i|z), \tag{32}$$

where $\hat{w}'_t(x^i|z)$, is the posterior of the scheduler $\kappa'_t$, $t' = \kappa^{-1}_{\kappa'_t}$, and $\kappa^{-1}$ is the inverse of $\kappa$. The scheduler change formula in equation 32 is proved in Proposition 8. We note that by Proposition 6, for mask modeling, i.e., $p = \delta_{\mathrm{m}}$, the posterior $\hat{w}_t(x^i|z)$ is time independent. Hence, in that case, the posterior is not affected by a scheduler change.

## C   Code infilling

We additionally evaluate the proposed method considering the task of code infilling. In which, we are provided with an input prompt that contains various spans of masked tokens, and our goal is to predict them based on the unmasked ones. See Figure 1 (middle and right sub-figures) for a visual example. Notice, this evaluation setup is the most similar to the training process.

For that, we randomly mask tokens with respect to several masking rations, $p \in \{0.0, 0.1, 0.2, \ldots, 1.0\}$, from HumanEval and report both pass@1 and compiles@1 metrics. For the purpose of this analysis, we provide the oracle length for each masked span. In other words, the model predicts the masked tokens for already given maks length. Results for the 1.5B parameters models can be seen in Figure 4. As expected, both pass@1 and compiles@1 keep

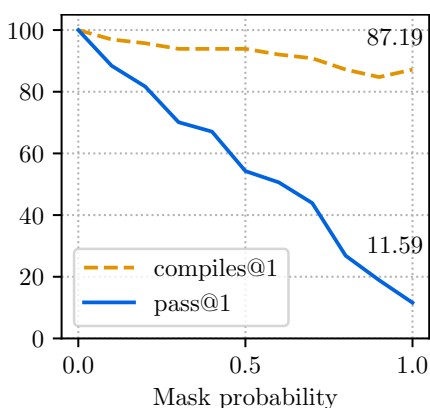

Figure 4: Pass@1 and compiles@1 scores for the 1.5B parameter models as a function of the input masking rations on HumanEval.

improving as we decrease the level of input masking. Interestingly, when considering the fully masked sequence, providing the oracle prediction length significantly improves the pass@1 scores (6.7 vs. 11.6).

## D   Ablations

**Train and sampling path scheduler choice ($\kappa_t$).**  We study how the choice of the probability path scheduler affects the model performance. For that, we consider a parametric family of cubic

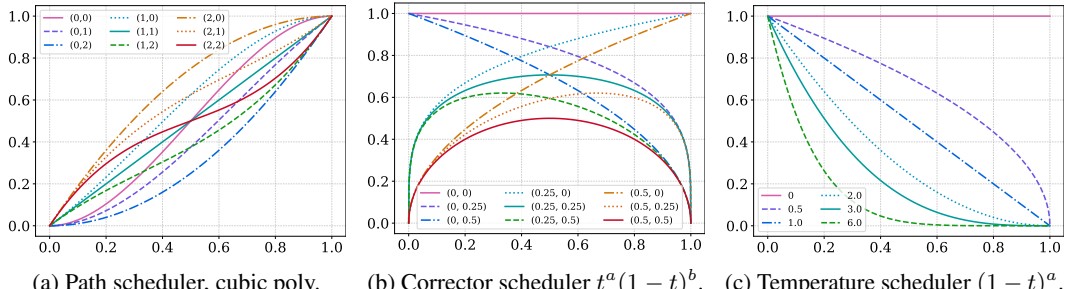

(a) Path scheduler, cubic poly.    (b) Corrector scheduler $t^a(1-t)^b$.    (c) Temperature scheduler $(1-t)^a$.

polynomial with parameters $a, b$:

$$\kappa_t \triangleq -2t^3 + 3t^2 + a(t^3 - 2t^2 + t) + b(t^3 - t^2). \tag{33}$$

Note that $\kappa_0 = 0$ and $\kappa_1 = 0$ and $a$ and $b$ are setting the derivative of $\kappa_t$ at $t = 0$ and $t = 1$, respectively. We visualize this $\kappa_t$ with choices of $a, b \in \{0, 1, 2\}$ in Figure 5a.

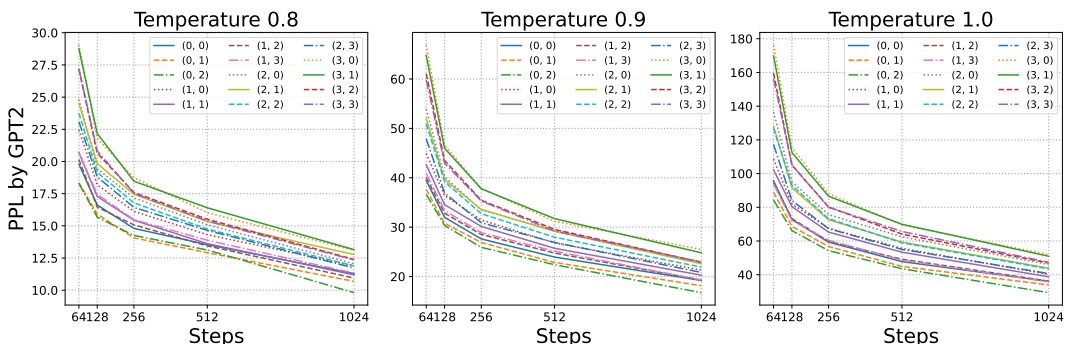

Figure 6: Path scheduler choice during training using various of constant temperature values.

To test the effect of path schedulers in training we have trained 150M parameters models for all choices of $a, b \in \{0, 1, 2, 3\}$. We then generate 1000 samples from each model. The samples are computed using Algorithm 1 with the path scheduler the model was trained on, and with temperature levels $\tau \in \{0.8, 0.9, 1\}$, where temperature is applied via

$$p_{1|t}^\tau(x^i|X_t) = \tau^{-1} \log p_{1|t}(x^i|X_t). \tag{34}$$

We then evaluate the generative perplexity of these samples with GPT-2. Figure 6 shows the results. The graphs indicate that, in the context of text modality, the cubic polynomial scheduler with $a \equiv 0, b \equiv 2$ (equivalent to a square function) achieves the highest performance. Consequently, we exclusively used this scheduler for the language models.

**Corrector scheduler.** In our experiments we only applied corrector sampling to our large models (U-coupling and C-coupling; 1.7B parameters). We used the optimal path schedulers from previous section and considered the following parametric family of schedulers for the corrector sampling:

$$\alpha_t = 1 + \alpha t^a(1-t)^b, \tag{35}$$

where, we set $\beta_t = \alpha_t - 1$ and generate 1000 samples using Algorithm 1 with parameter values $a, b \in \{0, 0.25, 0.5\}$ and $\alpha \in \{10, 15, 20\}$. We then evaluated generative perplexity for these samples with Llama-2, showing results in Figure 7. These plots indicate that smaller values of $a$ and $b$ result in lower perplexity values, albeit with somewhat reduced entropy. We therefore opted for setting $a = b = 0.25$ that strikes a good balance between perplexity and entropy.

**Temperature scheduling.** For temperature sampling, we consider the following scheduler:

$$\tau_t = \tau(1-t)^2. \tag{36}$$

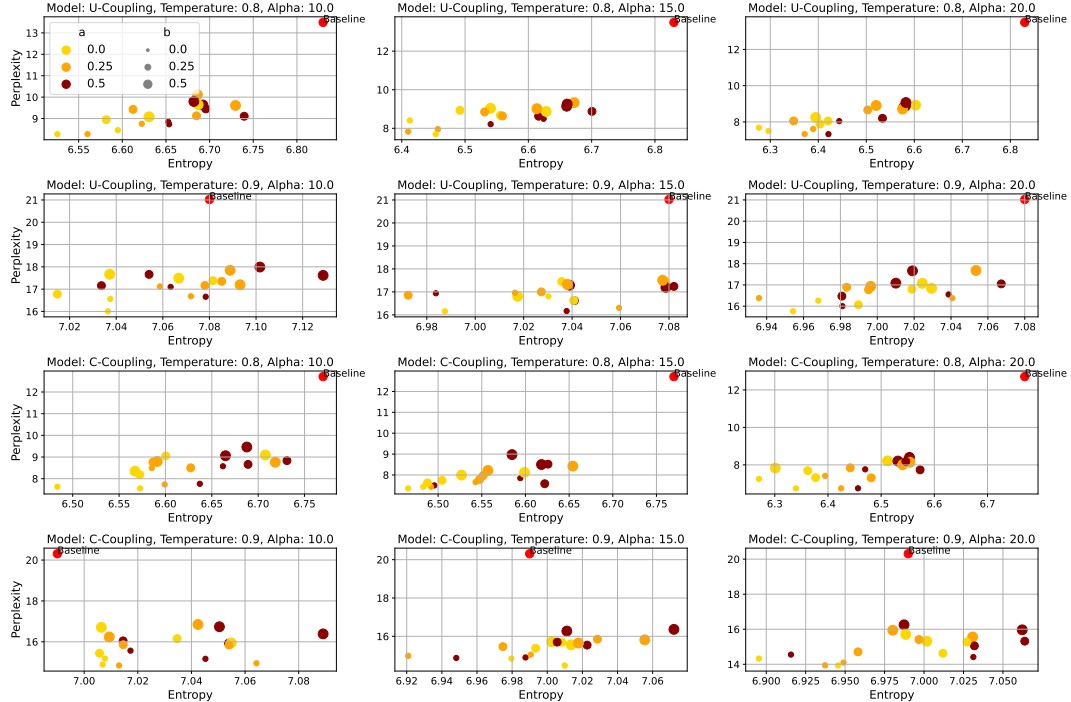

Figure 7: Corrector scheduler ablation.

# E Theory and proofs

## E.1 Computation of the discrete divergence

We present the computation of the discrete divergence in equation 18, i.e.,

$$\text{div}_x(p_t u_t) = -\sum_{z \in \mathcal{D}} p_t(z) \left[ \sum_{i=1}^{N} \delta_z(x^{\bar{i}}) u_t^i(x^i, z) \right]. \tag{37}$$

Computing the discrete divergence (equation 18) of the flux $p_t u_t$ at a state $x$ amounts to adding outgoing flux from $x$ and subtracting the incoming flux into $x$. Using the fact that $\delta_z(x^{\bar{i}}) = 1$ if and only if $z = x$ or $z$ differs from $x$ only at the $i$-th token, gives:

$$\text{div}_x(p_t u_t) = \sum_{z \in \mathcal{D}} \sum_{i=1}^{N} \delta_x(z^{\bar{i}}) \left( p_t(x) u_t^i(z^i, x) - p_t(z) u_t^i(x^i, z) \right)$$

$$= p_t(x) \sum_{i=1}^{N} \sum_{z^i} \overbrace{\left[ \sum_{z^{\bar{i}}} \delta_x(z^{\bar{i}}) \right]}^{=1} u_t^i(z^i, x) - \sum_{z \in \mathcal{D}} \sum_{i=1}^{N} \delta_x(z^{\bar{i}}) p_t(z) u_t^i(x^i, z)$$

$$= p_t(x) \sum_{i=1}^{N} \overbrace{\left[ \sum_{z^i} u_t^i(z^i, x) \right]}^{=0} - \sum_{z \in \mathcal{D}} \sum_{i=1}^{N} \delta_x(z^{\bar{i}}) p_t(z) u_t^i(x^i, z) \qquad \triangleright \text{ equation 13}$$

$$= -\sum_{z \in \mathcal{D}} \sum_{i=1}^{N} \delta_x(z^{\bar{i}}) p_t(z) u_t^i(x^i, z),$$

that gives equation 37 after noting that $\delta_x(z^{\bar{i}}) = \delta_z(x^{\bar{i}})$.

## E.2 Conditional velocities lead to marginal velocities

We provide a simple proof for Theorem 2, originally proved in Campbell et al. (2024):

**Theorem 2.** Given a conditional probability velocity $u_t^i(x^i, X_t | x_0, x_1)$ generating a conditional probability path $p_t(x | x_0, x_1)$, the marginal velocity defined by

$$u_t^i(x^i, X_t) = \sum_{x_0, x_1 \in \mathcal{D}} u_t^i(x^i, X_t | x_0, x_1) p_t(x_0, x_1 | X_t), \tag{38}$$

generates the marginal probability path $p_t(x)$, where by Bayes' rule

$$p_t(x_0, x_1 | X_t) = \frac{p_t(X_t | x_0, x_1) \pi(x_0, x_1)}{p_t(x)}. \tag{39}$$

*Proof (Theorem 2).* We start by taking the time derivative of the marginal probability path, $p_t(x) = \sum_{x_0, x_1} p_t(x^i | x_0, x_1) \pi(x_0, x_1)$, as follows,

$$
\begin{aligned}
\dot{p}_t(x) &= \sum_{x_0, x_1} \dot{p}_t(x | x_0, x_1) \pi(x_0, x_1) \\
&= \sum_{x_0, x_1} \left( \sum_z p_t(z | x_0, x_1) \left[ \sum_{i=1}^N \delta_z(x^{\bar{i}}) u_t^i(x^i, z | x_0, x_1) \right] \right) \pi(x_0, x_1) \qquad \triangleright \text{Continuity Equation (16)} \\
&= \sum_z p_t(z) \left[ \sum_{i=1}^N \delta_z(x^{\bar{i}}) \left( \sum_{x_0, x_1} u_t^i(x^i, z | x_0, x_1) \frac{p_t(z | x_0, x_1) \pi(x_0, x_1)}{p_t(z)} \right) \right] \\
&= \sum_z p_t(z) \left[ \sum_{i=1}^N \delta_z(x^{\bar{i}}) u_t^i(x^i, z) \right] \\
&= -\text{div}_x(p_t u_t)
\end{aligned}
$$

Now since $u_t^i(x^i, z)$ is a convex combinations of $u_t^i(x^i, z | x_0, x_1)$ and these satisfy equation 13 then also $u_t^i(x^i, X_t)$ satisfies equation 13. $\qquad \square$

## E.3 Probability velocities generating conditional probability paths

Equation 22 with the coefficients $a_t^{i,j}$ and $b_t^i$ are provided below,

$$u_t^i(x^i, X_t | x_0, x_1) = \sum_{j=1}^m \overbrace{\left[ \dot{\kappa}_t^{i,j} - \kappa_t^{i,j} \frac{\dot{\kappa}_t^{i,\ell}}{\kappa_t^{i,\ell}} \right]}^{a_t^{i,j}} w^j(x^i | x_0, x_1) + \overbrace{\left[ \frac{\dot{\kappa}_t^{i,\ell}}{\kappa_t^{i,\ell}} \right]}^{b_t^i} \delta_{X_t}(x^i), \tag{40}$$

where

$$\ell = \ell(i, t) \stackrel{\text{def}}{=} \arg\min_{j \in [m]} \left[ \dot{\kappa}_t^{i,j} / \kappa_t^{i,j} \right]. \tag{41}$$

**Theorem 3** (Probability velocity of conditional paths). A generating probability velocity for the conditional paths $p_t(x | x_0, x_1)$ defined in equations 7 and 8 is

$$u_t^i(x^i, X_t | x_0, x_1) = \sum_{j=1}^m a_t^{i,j} w^j(x^i | x_0, x_1) + b_t^i \delta_{X_t}(x^i), \tag{42}$$

with $a_t^{i,j} = \dot{\kappa}_t^{i,j} - \kappa_t^{i,j} \dot{\kappa}_t^{i,\ell} / \kappa_t^{i,\ell}$, and $b_t^i = \dot{\kappa}_t^{i,\ell} / \kappa_t^{i,\ell}$ where $\ell = \arg\min_{j \in [m]} \left[ \dot{\kappa}_t^{i,j} / \kappa_t^{i,j} \right]$.

*Proof (Theorem 3).* First, let us show that equation 40 satisfies the conditions in equation 13: Fix $X_t \in \mathcal{D}$, and

$$\sum_{x^i} u_t^i(x^i, X_t | x_0, x_1) = \sum_{x^i} \sum_{j=1}^{m} \left[ \dot{\kappa}_t^{i,j} - \kappa_t^{i,j} \frac{\dot{\kappa}_t^{i,\ell}}{\kappa_t^{i,\ell}} \right] w^j(x^i | x_0, x_1) + \frac{\dot{\kappa}_t^{i,\ell}}{\kappa_t^{i,\ell}} \delta_{X_t}(x^i)$$

$$= \sum_{j=1}^{m} \left[ \dot{\kappa}_t^{i,j} - \kappa_t^{i,j} \frac{\dot{\kappa}_t^{i,\ell}}{\kappa_t^{i,\ell}} \right] + \frac{\dot{\kappa}_t^{i,\ell}}{\kappa_t^{i,\ell}}$$

$$= \sum_{j=1}^{m} \dot{\kappa}_t^{i,j} + \frac{\dot{\kappa}_t^{i,\ell}}{\kappa_t^{i,\ell}} \left( 1 - \sum_{j=1}^{m} \kappa_t^{i,j} \right) \qquad \triangleright \sum_j \kappa_t^{i,j} = 1, \text{ and } \sum_j \dot{\kappa}_t^{i,j} = 0$$

$$= 0.$$

and for $x^i \neq X_t^i$ we have

$$u_t^i(x^i, X_t | x_0, x_1) = \sum_{j=1}^{m} \left[ \frac{\dot{\kappa}_t^{i,j}}{\kappa_t^{i,j}} - \frac{\dot{\kappa}_t^{i,\ell}}{\kappa_t^{i,\ell}} \right] \kappa_t^{i,j} w^j(x^i | x_0, x_1) \geq 0 \qquad (43)$$

since $\kappa_t^{i,j} \geq 0$, $\hat{w}_t(x^i | z) \geq 0$, and $\frac{\dot{\kappa}_t^{i,j}}{\kappa_t^{i,j}} - \frac{\dot{\kappa}_t^{i,\ell}}{\kappa_t^{i,\ell}} \geq 0$ since $\ell = \arg\min_{j \in [m]} \frac{\dot{\kappa}_t^{i,j}}{\kappa_t^{i,j}}$. Second, we show that $u_t$ satisfies the Continuity Equation (equation 16). To that end we write equation 8 as

$$w^\ell(x^i | x_0, x_1) = \frac{1}{\kappa_t^{i,\ell}} \left[ p_t(x^i | x_0, x_1) - \sum_{j \neq \ell} \kappa_t^{i,j} w^j(x^i | x_0, x_1) \right], \qquad (44)$$

where $\ell = \arg\min_{j \in [m]} \frac{\dot{\kappa}_t^{i,j}}{\kappa_t^{i,j}}$. Now by differentiating $p_t(x | x_0, x_1)$ we get

$$p_t(x | x_0, x_1) = \prod_{i=1}^{N} p_t(x^i | x_0, x_1)$$

$$\dot{p}_t(x | x_0, x_1) = \sum_{i=1}^{N} p_t(x^{\bar{i}} | x_0, x_1) \dot{p}_t(x^i | x_0, x_1)$$

$$= \sum_{i=1}^{N} p_t(x^{\bar{i}} | x_0, x_1) \left[ \sum_{j=1}^{m} \dot{\kappa}_t^{i,j} w^j(x^i | x_0, x_1) \right]$$

$$= \sum_{i=1}^{N} p_t(x^{\bar{i}} | x_0, x_1) \left[ \sum_{j \neq \ell} \dot{\kappa}_t^{i,j} w^j(x^i | x_0, x_1) + \dot{\kappa}_t^{i,\ell} w^\ell(x^i | x_0, x_1) \right]$$

$$= \sum_{i=1}^{N} p_t(x^{\bar{i}} | x_0, x_1) \left[ \sum_{j=1}^{m} \overbrace{\left[ \dot{\kappa}_t^{i,j} - \kappa_t^{i,j} \frac{\dot{\kappa}_t^{i,\ell}}{\kappa_t^{i,\ell}} \right]}^{a_t^{i,j}} w^j(x^i | x_0, x_1) + \overbrace{\frac{\dot{\kappa}_t^{i,\ell}}{\kappa_t^{i,\ell}}}^{b_t^i} p_t(x^i | x_0, x_1) \right] \qquad \triangleright \text{ equation 44}$$

$$= \sum_{i=1}^{N} \left[ \sum_{j=1}^{m} a_t^{i,j} \overbrace{\left[ \sum_z \delta_x(z^{\bar{i}}) p_t(z | x_0, x_1) \right]}^{= p_t(x^{\bar{i}} | x_0, x_1)} w^j(x^i | x_0, x_1) + b_t^i \overbrace{\left[ \sum_z \delta_x(z^{\bar{i}}) \delta_x(z^i) p_t(z | x_0, x_1) \right]}^{= p_t(x | x_0, x_1)} \right]$$

$$= \sum_z p_t(z | x_0, x_1) \sum_{i=1}^{N} \delta_x(z^{\bar{i}}) \overbrace{\left[ \sum_{j=1}^{m} a_t^{i,j} w^j(x^i | x_0, x_1) + b_t^i \delta_x(z^i) \right]}^{u_t^i(x^i, z | x_0, x_1)} \qquad \triangleright \delta_x(z^i) = \delta_z(x^i), \, \delta_x(z^{\bar{i}}) = \delta_z(x^{\bar{i}})$$

$$= -\text{div}_x(p_t(\cdot | x_0, x_1) u_t(\cdot | x_0, x_1)),$$

as required. □

## E.4 Backward-time generating probability velocity.

Here we prove the equivalent of Theorem 3 for backward-time generating probability field. But first, let us justify the backward sampling formula,

$$X^i_{t-h} \sim \delta_{X^i_t}(\cdot) - h u^i_t(\cdot, X_t). \tag{45}$$

Similar to equation 20 we have

$$\mathbb{E}_{X_t} \prod_{i=1}^N \left[ \delta_{X_t}(x^i) - h u^i_t(x^i, X_t) \right] = \mathbb{E}_{X_t} \left[ \delta_{X_t}(x) - h \sum_{i=1}^N \delta_{X_t}(x^{\bar{i}}) u^i_t(x^i, X_t) \right] + o(h)$$

$$= p_t(x) + h \mathrm{div}_x(p_t u_t) + o(h) \overset{(16)}{=} p_t(x) - h \dot{p}_t(x) + o(h) = p_{t-h}(x) + o(h).$$

Therefore if the Continuity equation holds and $-u_t$ satisfies the conditions in equation 13 then given $X_t \sim p_t$, equation 45 provides an approximation $X_{t-h} \sim p_{t-h} + o(h)$. The change to the generating probability velocity in equation 22 to accommodate reverse time sampling is to replace the argmin in equation 41 with argmax,

$$\ell = \ell(i, t) \triangleq \underset{j \in [m]}{\arg \max} \left[ \dot{\kappa}^{i,j}_t / \kappa^{i,j}_t \right]. \tag{46}$$

An analogous result to Theorem 3 for backward-time sampling is therefore,

**Theorem 7** (Probability velocity of conditional paths, backward time). *The probability velocity* $-u_t$, *where* $u_t$ *defined in equation 21 with* $\ell = \arg \max_{j \in [m]} \left[ \dot{\kappa}^{i,j}_t / \kappa^{i,j}_t \right]$ *is a backward-time generating probability velocity for the conditional paths* $p_t(x|x_0, x_1)$ *defined in equations 7 and 8.*

*Proof (Theorem 7).* We follow the proof of Theorem 3 and indicate the relevant changes. First, for arbitrary $X_t \in \mathcal{D}$,

$$\sum_{x^i} u^i_t(x^i, X_t) = 0, \tag{47}$$

exactly using the same arguments as the forward-time case. Now, for $x^i \neq X^i_t$ we have

$$u^i_t(x^i, X_t|x_0, x_1) = \sum_{j=1}^m \left[ \frac{\dot{\kappa}^{i,j}_t}{\kappa^{i,j}_t} - \frac{\dot{\kappa}^{i,\ell}_t}{\kappa^{i,\ell}_t} \right] \kappa^{i,j}_t w^j_t(x^i|x_0, x_1) \leq 0 \tag{48}$$

due to $\ell$ being now the argmax of $\frac{\dot{\kappa}^{i,j}_t}{\kappa^{i,j}_t}$. Therefore $-u_t$ satisfies equation 13. Lastly, we notice that the proof of the Continuity Equation follows through exactly the same also in this case. □

## E.5 Backward-time generating velocity for i.i.d. source $p(x_0)$ and simple paths

Here we consider the case of probability paths defined via the conditionals in equation 9 with independent coupling $\pi(x_0, x_1) = p(x_0)q(x_1)$ and i.i.d. source distribution $p(x_0) = \prod_{i=1}^N p(x^i_0)$, where $p(x^i_0)$ is some PMF over $[d]$. In this case one can simplify the time-backward sampling formula in equation 25 by using the following one which is equivalent (i.e., their difference is divergence free and consequently generate the same probability path $p_t$):

$$\check{u}_t(x^i, X_t) = \frac{\dot{\kappa}_t}{\kappa_t} \left[ \delta_{X_t}(x^i) - p(x^i) \right]. \tag{49}$$

The benefit in this equation is that it does not require the posterior $p_{0|t}$, which needs to be learned in general cases.

To show that equation 49 is indeed a generating probability velocity it is enough to show that

$$\mathrm{div}_x \left[ p_t \left( \check{u}_t - \check{u}^\star_t \right) \right] = 0, \tag{50}$$

where $\check{u}_t^\star$ is the probability velocity in equation 25. Let us verify using equation 19:

$$\text{div}_x\left[p_t\left(\check{u}_t - \check{u}_t^\star\right)\right] = \sum_{i,z} p_t(z)\delta_z(x^{\bar{i}})\left[p(x^i) - \sum_{x_0,x_1}\delta_{x_0}(x^i)\frac{p_t(z|x_0,x_1)p(x_0)q(x_1)}{p_t(z)}\right] \quad \triangleright \pi(x_0,x_1) = p(x_0)q(x_1)$$

$$= \sum_{i,z}\delta_z(x^{\bar{i}})\left[p(x^i)p_t(z) - \sum_{x_0,x_1}\delta_{x_0}(x^i)p_t(z|x_0,x_1)p(x_0)q(x_1)\right]$$

$$= \sum_{i,x_0,x_1}\left[p(x^i) - \delta_{x_0}(x^i)\right]\left(\sum_z\delta_z(x^{\bar{i}})p_t(z|x_0,x_1)\right)p(x_0)q(x_1)$$

$$= \sum_{i,x_0,x_1}\left[p(x^i) - \delta_{x_0}(x^i)\right]p_t(x^{\bar{i}}|x_0,x_1)p(x_0^i)p(x_0^{\bar{i}})q(x_1)$$

$$= \sum_{i,x_0^{\bar{i}},x_1}\left(\sum_{x_0^i}\left[p(x^i)p(x_0^i) - \delta_{x_0}(x^i)p(x_0^i)\right]\right)p_t(x^{\bar{i}}|x_0,x_1)p(x_0^{\bar{i}})q(x_1)$$

$$= 0,$$

where in the second to last equality we used the fact that the paths we are considering have the form:
$p_t(x^{\bar{i}}|x_0,x_1) = \prod_{j\in[N]\setminus i}\left[\kappa_t\delta_{x_1}(x^j) + (1-\kappa_t)\delta_{x_0}(x^j)\right]$, and therefore do not depend on the $i$-th source token, $x_0^i$.

### E.6 Corrector steps

***Theorem 4.*** For perfectly trained posteriors and $\alpha_t, \beta_t > 0$, $t \in (0,1)$, $\bar{u}_t$ in equation 26 is a probability velocity, i.e., satisfies equation 13, and: (i) For $\alpha_t - \beta_t = 1$, $\bar{u}_t$ provides a probability velocity generating $p_t$; (ii) For $\alpha_t - \beta_t = 0$, repeatedly sampling with $\bar{u}_t$ at fixed $t \in (0,1)$ and sufficiently small $h$ is guaranteed to converge to a sample from $p_t$.

*Proof (Theorem 4).* First let us write explicitly $\bar{u}_t$ from equation 26:

$$\bar{u}_t^i(x^i, X_t) = \alpha_t\hat{u}_t^i(x^i, X_t) - \beta_t\check{u}_t^i(x^i, X_t)$$

$$= \frac{\alpha_t\dot{\kappa}_t}{1-\kappa_t}p_{1|t}(x^i|X_t) + \frac{\beta_t\dot{\kappa}_t}{\kappa_t}p_{0|1}(x^i|X_t) - \left[\frac{\alpha_t\dot{\kappa}_t}{1-\kappa_t} + \frac{\beta_t\dot{\kappa}_t}{\kappa_t}\right]\delta_{X_t}(x^i). \quad (51)$$

Since equation 51 is a sum of PMFs with coefficients that sum up to zero the first condition in equation 13, i.e., $\sum_{x^i}\bar{u}_t^i(x^i, X_t) = 0$ holds. The second condition in equation 13 holds since for $t \in (0,1)$ we have $\frac{\alpha_t\dot{\kappa}_t}{1-\kappa_t}, \frac{\beta_t\dot{\kappa}_t}{\kappa_t} \geq 0$. Now,

$$\begin{aligned}\text{div}_x(p_t\bar{u}_t) &= \alpha_t\text{div}_x(p_t\hat{u}_t) - \beta_t\text{div}_x(p_t\check{u}_t) &&\triangleright \text{ linearity of div}\\ &= -\alpha_t\dot{p}_t(x) + \beta_t\dot{p}_t(x) &&\triangleright \text{ Equation 16}\\ &= -(\alpha_t - \beta_t)\dot{p}_t(x). && (52)\end{aligned}$$

**For (i):** Using equation 52 with $\alpha_t - \beta_t = 1$ we get that

$$\text{div}_x(p_t\bar{u}_t) = -\dot{p}_t(x),$$

i.e., $\bar{u}_t$ satisfies the continuity equation and therefore generates $p_t$.

**For (ii):** Setting $\alpha_t - \beta_t = 0$ in equation 52 we get $\text{div}_x(p_t\bar{u}_t) = 0$ and therefore similar to equation 20 we have

$$\begin{aligned}p_t(x) &= p_t(x) - h\text{div}_x(p_t\bar{u}_t)\\ &= \mathbb{E}_{X_t}\left[\delta_{X_t}(x) + h\sum_{i=1}^N\delta_{X_t}(x^{\bar{i}})\bar{u}_t^i(x^i, X_t)\right]\\ &= \sum_z p(x|z)p_t(z), && (53)\end{aligned}$$

where using equation 51 we have

$$p(x|z) = h \sum_{i=1}^{N} \frac{\alpha_t \dot{\kappa}_t}{1 - \kappa_t} \delta_z(x^{\bar{i}}) p_{1|t}(x^i|z) + h \sum_{i=1}^{N} \frac{\beta_t \dot{\kappa}_t}{\kappa_t} \delta_z(x^{\bar{i}}) p_{0|1}(x^i|z)$$

$$+ \left(1 - h \sum_{i=1}^{N} \left[ \frac{\alpha_t \dot{\kappa}_t}{1 - \kappa_t} + \frac{\beta_t \dot{\kappa}_t}{\kappa_t} \right] \right) \delta_z(x^i).$$

For sufficiently small $h > 0$ therefore $p(x|z)$ is a convex combination of PMFs (in red) $x$ and consequently is itself a PMF in $x$, that is $p(x|z)$ is a probability transition matrix, and $p_t(x)$ is its stationary distribution, i.e., it is an eigenvector of $p(x|z)$ with eigenvalue 1, which is maximal. To prove convergence of the iterations in equation 53 we are left with showing that $p(x|z)$ is irreducible and a-periodic, see Norris (1998) (Theorem 1.8.3). Irreducibly of $p(x|z)$ can be shown by connecting each two states $z, z'$ by changing one token at a time, and assuming that $p_{1|t}$ or $p_{0|t}$ are strictly positive (which is usually the case since as at-least one of them is defined as soft-max of finite logits); a-periodicity is proved by showing $p(x|x) > 0$ which is true as the coefficient of $\delta_z(x)$ is greater than zero for sufficiently small $h > 0$. Lastly, note that the iteration in equation 53 changes one token at a time. An approximation to this sampling can be achieved using our standard parallel sampling via equation 12, justified by equation 20.

□

## E.7 Training

**Proposition 5.** The minimizer of $\mathcal{L}$ (equation 28) is $\hat{w}_t^j(x^i|X_t)$ (equation 23).

*Proof (Proposition 5).* It is enough to prove the claim for $m = 1$, with a single $w(x^i|x_0, x_1)$.

$$\mathcal{L}(\theta) = -\frac{1}{N} \sum_{i=1}^{N} \mathbb{E}_t \sum_{x_0, x_1, z, y^i} \log \hat{w}_t(y^i|z; \theta) w(y^i|x_0, x_1) p_t(z|x_0, x_1) \pi(x_0, x_1)$$

$$= -\frac{1}{N} \sum_{i=1}^{N} \mathbb{E}_t \sum_z p_t(z) \left[ \sum_{y^i} \log \hat{w}_t(y^i|z; \theta) \left( \sum_{x_0, x_1} w(y^i|x_0, x_1) \frac{p_t(z|x_0, x_1) \pi(x_0, x_1)}{p_t(z)} \right) \right]$$

$$= -\mathbb{E}_{t, X_t} \frac{1}{N} \sum_{i=1}^{N} \left[ \sum_{y^i} \log \hat{w}_t(y^i|X_t; \theta) \hat{w}_t(y^i|X_t) \right],$$

that amounts to minimizing the Cross Entropy loss between $\hat{w}_t(x^i|X_t; \theta)$ and $\hat{w}_t(x^i|X_t)$ for all $i \in [N]$, the minimizer of which satisfies $\hat{w}_t(x^i|X_t; \theta) \equiv \hat{w}_t(x^i|X_t)$. □

## E.8 Time-independent posterior for masked modeling

**Proposition 6.** For paths defined by equations 7 and 9 with source $p(x) = \delta_m(x)$ the posterior $p_t(x_0, x_1|z) = p(x_0, x_1|z)$ is time-independent. Consequently, the probability denoiser $p_{1|t}(x^i|z) = p_1(x^i|z)$ is also time-independent.

*Proof (Proposition 6).* First,

$$p_t(z^i|x_0, x_1) = (1 - \kappa_t) \delta_m(z^i) + \kappa_t \delta_{x_1}(z^i) = \begin{cases} (1 - \kappa_t) & z^i = m \\ \kappa_t \delta_{x_1}(z^i) & z^i \neq m \end{cases}$$

and therefore

$$p_t(z|x_0, x_1) = \left[ \prod_{i:z^i=m} (1 - \kappa_t) \prod_{i:z^i \neq m} \kappa_t \right] \prod_{i:z^i \neq m} \delta_{x_1}(z^i).$$

The posterior now gives

$$
\frac{p_t(z|x_0,x_1)\pi(x_0,x_1)}{p_t(z)} = \frac{\left[\prod_{i=1}^N p_t(z^i|x_0,x_1)\right]\pi(x_0,x_1)}{\sum_{\tilde{x}_0,\tilde{x}_1}\left[\prod_{j=1}^N p_t(z^j|\tilde{x}_0,\tilde{x}_1)\right]\pi(\tilde{x}_0,\tilde{x}_1)}
$$

$$
= \frac{\left[\cancel{\prod_{i:z^i=\mathbb{m}}(1-\kappa_t)}\prod_{i:z^i\neq\mathbb{m}}\kappa_t\right]\left[\prod_{i:z^i\neq\mathbb{m}}\delta_{x_1}(z^i)\right]\pi(x_0,x_1)}{\sum_{\tilde{x}_0,\tilde{x}_1}\left[\cancel{\prod_{j:z^j=\mathbb{m}}(1-\kappa_t)}\prod_{j:z^j\neq\mathbb{m}}\kappa_t\right]\left[\prod_{j:z^j\neq\mathbb{m}}\delta_{\tilde{x}_1}(z^j)\right]\pi(\tilde{x}_0,\tilde{x}_1)}
$$

$$
= p(x_0,x_1|z).
$$

showing that the posterior is time-independent for dummy source distributions and convex paths. Consequently also the probability denoiser,

$$
p_{1|t}(x^i|z) = \sum_{x_0,x_1}\delta_{x_1}(x^i)\frac{p_t(z|x_0,x_1)\pi(x_0,x_1)}{p_t(z)} = \sum_{x_0,x_1}\delta_{x_1}(x^i)p(x_0,x_1|z),
$$

is time-independent. $\qquad\square$

### E.9 Continuous Flow Matching

For completeness we provide the formulas for denoiser ($x$-prediction) and noise-prediction ($\varepsilon$-prediction) parameterizations of the generating velocity field $u : [0,1] \times \mathbb{R}^N \to \mathbb{R}^N$ appearing in Table 1.

In Continuous Flow Matching one can chose several ways to define the probability paths (Lipman et al., 2022; Liu et al., 2022; Albergo and Vanden-Eijnden, 2022; Pooladian et al., 2023; Tong et al., 2023):

$$
p_t(x) = \int p_t(x|x_0,x_1)\pi(x_0,x_1)dx_0dx_1 \tag{54}
$$

$$
= \int p_{1|t}(x|x_1)q(x_1)dx_1 \tag{55}
$$

$$
= \int p_{0|t}(x|x_0)p(x_0)dx_0. \tag{56}
$$

**Denoiser parameterization.** The conditional generating velocity field $u_t(x|x_1)$ for $p_t(x|x_1)$, i.e., satisfy the Continuity Equation 16, takes the form (Lipman et al., 2022)

$$
u_t(x|x_1) = \frac{\dot{\kappa}_t}{1-\kappa_t}(x_1 - x), \tag{57}
$$

and the marginal generating velocity field is therefore given by the marginalization with the posterior $p_t(x_1|x)$,

$$
u_t(x) = \int \frac{\dot{\kappa}_t}{1-\kappa_t}(x_1 - x)\frac{p_{1|t}(x|x_1)q(x_1)}{p_t(x)}dx_1
$$

$$
= \frac{\dot{\kappa}_t}{1-\kappa_t}\left[\hat{x}_{1|t}(x) - x\right],
$$

where

$$
\hat{x}_{1|t}(x) = \int x_1\frac{p_{1|t}(x|x_1)q(x_1)}{p_t(x)}dx_1 = \mathbb{E}_{X_1\sim p_t(\cdot|x)}X_1. \tag{58}
$$

This shows the continuous Flow Matching denoiser parameterization of the generating velocity field in Table 1.

**Noise-prediction parameterization.** The conditional generating velocity field for $p_t(x|x_0)$ takes the form

$$
u_t(x|x_0) = \frac{\dot{\kappa}_t}{\kappa_t}(x - x_0), \tag{59}
$$

and the marginal generating velocity field in this case is given by marginalization with the posterior $p_t(x_0|x)$,

$$
\begin{aligned}
u_t(x) &= \int \frac{\dot{\kappa}_t}{\kappa_t}(x - x_0) \frac{p_{0|t}(x|x_0)p(x_0)}{p_t(x)} dx_0 \\
&= \frac{\dot{\kappa}_t}{\kappa_t}\left[x - \hat{x}_{0|t}(x)\right],
\end{aligned}
$$

where

$$
\hat{x}_{0|t}(x) = \int x_0 \frac{p_{0|t}(x|x_0)p(x_0)}{p_t(x)} dx_0 = \mathbb{E}_{X_0 \sim p_t(\cdot|x)} X_0. \tag{60}
$$

This shows the continuous Flow Matching noise-prediction parameterization of the generating velocity field in Table 1.

### E.10 Scheduler change formula

**Proposition 8.** *Assume a conditional probability path as in equation 9, then for any two schedulers $\kappa_t$, $\kappa_t'$, and $\hat{w}_t(x^i|z)$, $\hat{w}_t'(x^i|z)$ their corresponding posteriors as in equation 23,*

$$
\hat{w}_{t'}(x^i|z) = \hat{w}_t'(x^i|z), \tag{61}
$$

*where $t' = \kappa^{-1}_{\kappa_t'}$, and $\kappa^{-1}$ is the inverse of $\kappa$.*

*Proof (Proposition 8).* For a conditional probability path as in equation 9,

$$
p_{t'}(x^i|x_0, x_1) = \prod_{i=1}^{N} p_{t'}(x^i|x_0, x_1) \tag{62}
$$

$$
= \prod_{i=1}^{N} \left[(1 - \kappa_{t'})\delta_{x_0}(x^i) + \kappa_{t'}\delta_{x_1}(x^i)\right] \tag{63}
$$

$$
= \prod_{i=1}^{N} \left[(1 - \kappa_t')\delta_{x_0}(x^i) + \kappa_t'\delta_{x_1}(x^i)\right] \tag{64}
$$

$$
= \prod_{i=1}^{N} p_t'(x^i|x_0, x_1) \tag{65}
$$

$$
= p_t'(x^i|x_0, x_1), \tag{66}
$$

where in the 3rd equality we used $\kappa_{t'} = \kappa_t'$. Thus, also for the marginal probability path as in equation 7,

$$
p_{t'}(x) = \sum_{x_0,x_1 \in \mathcal{D}} p_{t'}(x|x_0, x_1)\pi(x_0, x_1) \tag{67}
$$

$$
= \sum_{x_0,x_1 \in \mathcal{D}} p_t'(x|x_0, x_1)\pi(x_0, x_1) \tag{68}
$$

$$
= p_t'(x), \tag{69}
$$

where in the 2nd equality we used $p_{t'}(x|x_0, x_1) = p'_t(x|x_0, x_1)$. Finally the change of scheduler for a posterior as defined in equation 23,

$$\hat{w}_{t'}(x^i|z) = \sum_{x_0, x_1 \in \mathcal{D}} w(x^i|x_0, x_1) p_{t'}(x_0, x_1|z) \tag{70}$$

$$= \sum_{x_0, x_1 \in \mathcal{D}} w(x^i|x_0, x_1) \frac{p_{t'}(z|x_0, x_1)\pi(x_0, x_1)}{p_{t'}(z)} \tag{71}$$

$$= \sum_{x_0, x_1 \in \mathcal{D}} w(x^i|x_0, x_1) \frac{p'_t(z|x_0, x_1)\pi(x_0, x_1)}{p'_t(z)} \tag{72}$$

$$= \sum_{x_0, x_1 \in \mathcal{D}} w(x^i|x_0, x_1) p'_t(x_0, x_1|z) \tag{73}$$

$$= \hat{w}'_t(x^i|z) \tag{74}$$

where in the 3rd equality we used both $p_{t'}(z|x_0, x_1) = p'_t(z|x_0, x_1)$ and $p_{t'}(z) = p'_t(z)$. □

# F   Inference time

One potential benefit of non-autoregressive decoding is improved latency due to a significantly lower number of decoding steps. To demonstrate that, we measure the average latency of the proposed method compared with the autoregressive alternative using a single A100 GPU with 80 GB of RAM. We calculate the average latency time on the HumanEval benchmark using a batch size of 1. When considering 256 NFEs, the proposed method was found to be ~2.5x faster than the autoregressive model (19.97 vs. 50.94 seconds on average per example). However, when considering 512 NFEs, both methods reach roughly the same latency. These results make sense as the number of tokens in most of the examples in HumanEval are below 512. Notice, that these results analyze latency and not model throughput. Due to the kv-caching mechanism following the autoregressive approach will result in significantly better throughput compared to the proposed approach Ziv et al. (2024). We leave the construction of a kv-cache mechanism to the proposed approach for future research.

# G   Experimental setup

## G.1   Text

**Data.**   We use three splits of data. First is OpenWebText (Gokaslan and Cohen, 2019). Second is the same mix used in Llama-2 (Touvron et al., 2023), including textual and code data. For the code-focused models we use the same split used in CodeLlama (Roziere et al., 2023). For the small models, we use OpenWebText. For the big models we use the Llama-2 and CodeLlama mixes.

**Models.**   We train two sizes of models: small (150M parameters) and large (1.7B parameters). For the small model we used a DiT transformer architecture (Peebles and Xie, 2022) with 12 layers, 12 attention heads, and hidden dimension of 768. We also used GPT2 tokenizer. The small models were trained on OpenWebText. For the large model, we use also used a DiT transformer architecture but with 48 layers, 24 attention heads, and hidden dimension of 1536 (Peebles and Xie, 2022). For these models we used a tiktoken tokenizer. The large models were trained on the Llama-2 mix and the CodeLlama mix. For both models we used ROPE (Su et al., 2024) embedding with $\theta = 10000$. Models are trained with Adam optimizer with $\beta_1 = 0.9$ and $\beta_2 = 0.999$. We use dropout rate of 0.1. Models are trained with a warm-up of 2500 steps, with a peak learning rate of 3e-4. We train the big models with batch size of 4096 for 1.3 million iterations and the big models with batch size of 512 for 400 thousand iterations.

**Entropy metric.**   We report the entropy of tokens within a sequence, averaged over all generated sequences. This intuitively quantifies the diversity of tokens within a given sequence. It's important to note that when computing sequence entropy, tokens not present in the sequence are excluded from consideration.

**Generative perplexity metric.** The generative perplexity metric is the average likelihood of generated text evaluated with a second (usually stronger) model. We report the generative perplexity when averaged over 1000 samples.

**Double precision sampling.** Zheng et al. (2024) demonstrated that sampling from a high-dimensional distribution with full precision can lead to a similar affect as sampling with temperature. We evaluate our model using a categorical sampler in double precision. Table 5 presents the results of baselines compared to our method.

Table 5: **Double precision sampling.** Generative perplexity on unconditional text generation compared to prior work. All models are sampled without the use of temperature or corrector steps.

| METHOD | NFE | LLAMA-2↓ | LLAMA-3↓ | GPT2↓ | ENTROPY |
|---|---|---|---|---|---|
| Data | - | 7.0 | 9.4 | 14.7 | 7.7 |
| Autoregressive | 1024 | 31.4 | 54.8 | 45.3 | 7.1 |
| Savinov et al. (2021) | 200 | 29.5 | 45.1 | 34.7 | 5.2 |
| Austin et al. (2021a) | 1000 | 697.6 | 768.8 | 837.8 | 7.6 |
| Han et al. (2022) | >10000 | 73.3 | 203.1 | 99.2 | 4.8 |
| Lou et al. (2023) | 256/512/1024 | 56.6/54.0/56.1 | 122.1/115.7/117.7 | 115.0/107.8/109.5 | 8.1/8.1/8.1 |
| Campbell et al. (2024) | 256/512/1024 | 52.0/54.6/50.9 | 106.0/114.1/102.9 | 102.6/107.1/103.4 | 8.0/8.1/8.0 |
| **FM (equation 9)** | 256/512/1024 | 51.3/53.3/50.1 | 104.0/115.0/101.3 | 100.8/107.4/97.5 | 8.0/8.1/8.0 |
| **FM (equation 10)** | 256/512/1024 | 51.9/52.7/50.0 | 104.7/113.9/100.5 | 99.2/105.1/95.8 | 8.0/8.1/8.0 |

## G.2 Image

**Models.** For all our experiments on CIFAR10 we use the U-Net architecture as in Dhariwal and Nichol (2021), with following three changes to make it fully discrete and time independent (as we used mask modeling): (i) We replace the first layer with an embedding table of size $257 \times 96$, and we stack the channel features such that the input to the U-Net is of shape $288 \times 32 \times 32$. (ii) We enlarge the size of the final layer to output a tensor of shape $3 \times 32 \times 32 \times 257$. (iii) We remove the time dependency from architecture. The hyper-parameters of the architecture: channels 96 , depth 5, channels multiple [3,4,4], heads channels 64, attention resolution 16, dropout 0.4, which gives a total parameters count of 113M. We optimize the network using Adam optimizer with $\beta_1 = 0.9$ and $\beta_2 = 0.999$, a learning rate of 1e-4. We trained with an effective batch size pf 512 for roughly 300K iterations.

**Scheduler ablation.** Figure 8 shows FID of our method with four different schedulers: Linear, Quadratic, Cubic, Cosine, both for training and evaluation. That is, for each scheduler we trained a model and evaluate FID with all four schedulers. We observe a high variance in FID between different schedulers, with the Cubic scheduler generally performing the best on both training and evaluation.

**Comparison with baselines.** In the following, we provide implementation details for producing Figure 3a, that compares our schedulers and sampling algorithm with those employed by previous works.

**Cubic Scheduler (Ours).** For the Cubic scheduler we set the corrector scheduler as above to,

$$\alpha_t = 1 + \alpha t^a (1-t)^b, \quad \beta_t = \alpha_t - 1, \tag{75}$$

and we search over the parameters $a, b \in \{0, 0.25, 0.5, 1, 2, 2.5, 3\}$, and $\alpha \in \{6, 8, 10, 12, 14\}$. Additionally, we search over the temperature $\in \{1, 0.9, 0.8\}$. We find that $a = 2$, $b = 0.25$, $\alpha = 12$ give best FID.

**Linear Scheduler (Campbell et al., 2024).** For the linear scheduler we search over two additional hyper-parameters of the method: (i) For corrector scheduler as in equation 26, we set $\alpha_t = 1 + t\eta$, $\beta_t = \alpha_t - 1$, where $\eta$ is the stochasticity parameter as in Campbell et al. (2024), and search over $\eta \in \{0, 1, 2, 5, 10, 15\}$. (ii) We search over temperature in $\{1, 0.9, 0.8\}$. Finally, we find that the best FID is a achieved by $\eta = 10$ and temperature 0.9.

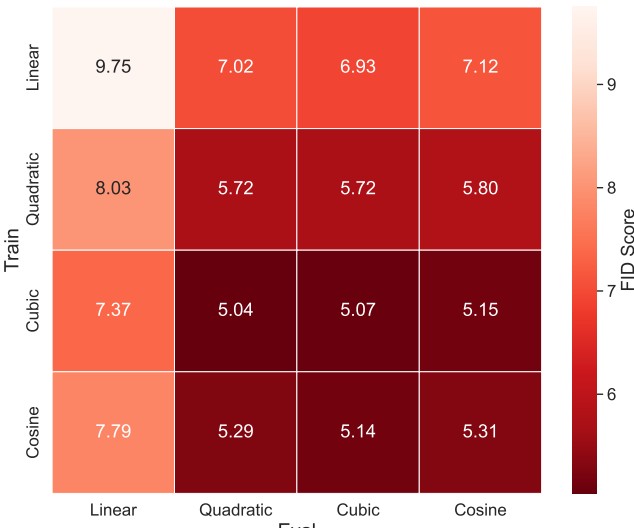

Figure 8: Comparison of FID on CIFAR10 with four schedulers: Linear, Quadratic, Cubic, Cosine, for both train and evaluation. Corrector sampling is not used in this experiment (i.e., $\alpha_t = 0$ in equation 26), and temperature is set to 1.

**MaskGIT (Chang et al., 2022).** For the MaskGIT we train and sample with the Cosine scheduler $\kappa(t) = 1 - \cos\left(\frac{\pi}{2}t\right)$ which is reported to achieve best results by Chang et al. (2022). For sampling we adjust the code from the re-implementation of Besnier and Chen (2023). In addition, we also search over the temperature in $\{1, 0.9, 0.8, 0.7, 0.6, 0.5\}$, and we find the best FID is achieved by temperature 1.

# H  Code generation - qualitative examples

## H.1  Success cases

```python
from typing import List

def below_zero(operations: List[int]) -> bool:
    """ You're given a list of deposit and withdrawal operations
    on a bank account that starts with
    zero balance. Your task is to detect if at any point the balance
    of account fallls below zero, and
    at that point function should return True.
    Otherwise it should return False.
    >>> below_zero([1, 2, 3])
    False
    >>> below_zero([1, 2, -4, 5])
    True
    """
```

```python
    balance = 0
    for operation in operations:
        if balance + operation < 0:
            return True
        balance = balance + operation
    return False
```

```python
def encode_shift(s: str):
    """
    returns encoded string by shifting every character by 5 in the alphabet.
    """
    return "".join([chr(((ord(ch) + 5 - ord("a")) % 26) + ord("a")) for ch in s])

def decode_shift(s: str):
    """
    takes as input string encoded with encode_shift function.
    Returns decoded string.
    """
```

```python
    decoded = []
    for ch in s:
        decoded.append(chr(((ord(ch) - 5 - ord("a")) % 26) + ord("a")))
    return "".join(decoded)
```

```python
from typing import List, Any

def filter_integers(values: List[Any]) -> List[int]:
    """ Filter given list of any python values only for integers
    >>> filter_integers(['a', 3.14, 5])
    [5]
    >>> filter_integers([1, 2, 3, 'abc', {}, []])
    [1, 2, 3]
    """

    return [value for value in values if isinstance(value, int)]
```

```python
def common(l1: list, l2: list):
    """Return sorted unique common elements for two lists.
    >>> common([1, 4, 3, 34, 653, 2, 5], [5, 7, 1, 5, 9, 653, 121])
    [1, 5, 653]
    >>> common([5, 3, 2, 8], [3, 2])
    [2, 3]

    """

    return sorted(set(l1) & set(l2))
```

```python
def fibfib(n: int):
    """The FibFib number sequence is a sequence similar to
    the Fibbonacci sequnece that's defined as follows:
    fibfib(0) == 0
    fibfib(1) == 0
    fibfib(2) == 1
    fibfib(n) == fibfib(n-1) + fibfib(n-2) + fibfib(n-3).
    Please write a function to efficiently compute the
    n-th element of the fibfib number sequence.
    >>> fibfib(1)
    0
    >>> fibfib(5)
    4
    >>> fibfib(8)
    24
    """

    if n == 0:
        return 0
    elif n == 1:
        return 0
    elif n == 2:
        return 1
    else:
        return fibfib(n-1) + fibfib(n-2) + fibfib(n-3)
```

## H.2 Failure cases

```python
def solve(s):
    """You are given a string s.
    if s[i] is a letter, reverse its case from lower to upper or vise versa,
    otherwise keep it as it is.
    If the string contains no letters, reverse the string.
    The function should return the resulted string.
    Examples
    solve("1234") = "4321"
    solve("ab") = "AB"
    solve("#a@C") = "#A@c"
    """

    if s is None:
        return ""
    if len(s) == 0:
        return s
    # result string
    res = str()
    # to lower case
    s = s.lower()
    for i in range(len(s)):
        ch = s[i]
        if ch.islower():
            res.append(ch)
        elif ch.isupper():
            res.append(ch.lower())
        else:
            res.append(ch)
    return res
```

```python
from typing import List, Tuple

def rolling_max(numbers: List[int]) -> List[int]:
    """ From a given list of integers, generate a list of rolling maximum element
    found until given moment in the sequence.
    >>> rolling_max([1, 2, 3, 2, 3, 4, 2])
    [1, 2, 3, 3, 3, 4, 4]
    """

    max_number = 0
    result = []
    for number in numbers:
        max_number = max(result)
        if number > max_number:
            result.append(number)
        else:
            result = [max_number]
    return result
```

```python
def truncate_number(number: float) -> float:
    """ Given a positive floating point number, it can be decomposed into
    and integer part (largest integer smaller than given number) and decimals
    (leftover part always smaller than 1).

    Return the decimal part of the number.
    >>> truncate_number(3.5)
    0.5
    """

    return math.floor(number)
```

```python
from typing import List, Tuple

def sum_product(numbers: List[int]) -> Tuple[int, int]:
    """ For a given list of integers, return a tuple consisting of a sum and a
    product of all the integers in a list. Empty sum should be equal to 0 and
    empty product should be equal to 1.
    >>> sum_product([])
    (0, 1)
    >>> sum_product([1, 2, 3, 4])
    (10, 24)
    """

    if len(numbers) == 0 or numbers[0] == 0:
        return 0, 1
    sum = 0
    product = 1
    for num in numbers:
        if abs(num) != 0:
            product = product * num
            sum += num
    return sum, product
```

```python
from typing import List

def factorize(n: int) -> List[int]:
    """ Return list of prime factors of given integer in the order from smallest
    to largest. Each of the factors should be listed number of times corresponding
    to how many times it appeares in factorization. Input number should be equal
    to the product of all factors.
    >>> factorize(8)
    [2, 2, 2]
    >>> factorize(25)
    [5, 5]
    >>> factorize(70)
    [2, 5, 7]
    """

    factors = []
    factor = 1
    for i in range(2, n):
        factor *= i
        while n % factor == 0:
            factors.append(factor)
            factor += 1
    return factors
```

```python
from typing import List, Optional

def longest(strings: List[str]) -> Optional[str]:
    """ Out of list of strings, return the longest one. Return the first one in
    case of multiple strings of the same length. Return None in case the input
    list is empty.
    >>> longest([])

    >>> longest(['a', 'b', 'c'])
    'a'
    >>> longest(['a', 'bb', 'ccc'])
    'ccc'
    """

    long = ""
    for string in strings:
        if len(string) > len(long):
            long = string
    return long
```

## H.3  Infilling

```python
def bubbleSort(arr):
    n = len(arr)
    # optimize code, so if the array is already sorted, it doesn't need
    # to go through the entire process
    # Traverse through all array elements
    for i in range(n-1):

        # range(n) also work but outer loop will
        # repeat one time more than needed.
        # Last i elements are already in place
        swapped = False
        for j in range(0, n-i-1):
            # traverse the array from 0 to n-i-1
            # Swap if the element found is greater
            # than the next element
            if arr[j] > arr[j + 1]:
                swapped = True
                arr[j], arr[j + 1] = arr[j + 1], arr[j]
        if not swapped:
            # if we haven't needed to make a single swap, we
            # can just exit the main loop.
            return
```

```python
# Function to perform Breadth First Search on a graph
# represented using adjacency list
def bfs(adjList,  source , visited):
    # Create a queue for BFS
    q = deque()

    # Mark the current node as visited and enqueue it
    visited[ source ] = True

    q.append( source )

    # Iterate over the queue
    while q:
        # Dequeue a vertex from queue and print it
        currentNode = q.popleft()
        print( currentNode , end=" ")

        # Get all adjacent vertices of the dequeued vertex
        # If an adjacent has not been visited, then mark it visited and enqueue it
        for  adjacent  in adjList[ currentNode ]:

            if not visited[ adjacent ]:

                visited[ adjacent ] = True

                q.append( adjacent )
```

```python
# Returns index of x in arr if present, else -1
def binary_search(arr, low , high , x):

    # Check base case
    if high >= low :

        mid = ( high + low ) // 2

        # If element is present at the middle itself
        if arr[ mid ] == x:
            return mid

        # If element is smaller than mid, then it can only
        # be present in left subarray
        elif arr[ mid ] > x:

            return binary_search(arr, low , mid - 1, x)

        # Else the element can only be present in right subarray
        else:
            return binary_search(arr, mid + 1, high , x)

    else:
        # Element is not present in the array
        return -1
```

```python
# Python program for Dijkstra's single
# source shortest path algorithm. The program is
# for adjacency matrix representation of the graph
class Graph():

    def __init__(self, vertices):
        self.V = vertices
        self.graph = [[0 for column in range(vertices)]
                      for row in range(vertices)]

    def printSolution(self, dist):
        print("Vertex        Distance from Source")
        for node in range(self.V):
            print(node, "                ", dist[node])

    # A utility function to find the vertex with
    # minimum distance value, from the set of vertices
    # not yet included in shortest path tree
    def minDistance(self, dist, sptSet):

        # Initialize minimum distance for next node
        min = 1e7

        # Search not nearest vertex not in the
        # shortest path tree
        for v in range(self.V):
            if dist[v] < min and sptSet[v] == False:
                min = dist[v]
                min_index = v

        return min_index

    # Function that implements Dijkstra's single source
    # shortest path algorithm for a graph represented
    # using adjacency matrix representation
    def dijkstra(self, src ):

        dist  = [1e7] * self.V

        dist [src] = 0

        processed  = [False] * self.V

        for cout in range(self.V):

            # Pick the minimum distance vertex from
            # the set of vertices not yet processed.
            # u is always equal to src in first iteration
            uv  = self.minDistance(dist,  processed )

            # Put the minimum distance vertex in the
            # shortest path tree
            processed [ uv ] = True

            # Update distance value of the adjacent vertices
            # of the picked vertex only if the current
            # distance is greater than new distance and
            # the vertex in not in the shortest path tree
            for  v  in range(self.V):
                if (self.graph[ uv ][ v ] > 0 and
                    processed [ uv ] == False and
                    dist [ uv ] >  dist [ cout ] + self.graph[ uv ][ v ]):
                     dist [ uv ] =  dist [ cout ] + self.graph[ uv ][ v ]

        self.printSolution(dist)
```

# I   Textual generations

We present below example generations for the proposed method together with several baseline methods. We provide both conditional and unconditional generations. For the conditional generations, we mark the prompt in gray.

## I.1   Conditional generation

> The United States on Wednesday asked the UN Security Council to slap an oil embargo on North Korea and freeze the assets of leader Kim Jong-un, in response to Pyongyang's

response to the revelations it had restarted its nuclear work in March. "We will continue working to use maximum international pressure on North Korea to agree to the suspension of its nuclear program and reinstate sanctions," said John Bolton, who served as National Security Advisor and Secretary of State under US President Bill Clinton. "Here is North Korea's response to our sanctions," Bolton wrote in a letter to House Minority Leader Nancy Pelosi. "We want you to know that the international community is seriously monitoring North Korea at this time. North Korea is still complying with our requests from the past few days," Bolton said on Monday. "We have been working through the United Nations to provide the information that they gave us." Asked to whether any international pressure will be put in place for North Korea to give up its nuclear weapons, Bolton said the United States can use maximum pressure to get North Korea to abandon its nuclear weapons if it wants. "We've been working to use maximum pressure on North Korea, through the Security Council, and we will continue to do so," said White House Deputy Press Secretary Sarah Huckabee Sanders in Washington. "We're committed to taking any steps necessary to help North Korea pursue its only option for peace, including in this period," she added. The United States did not plan to produce any more oil at this time last year and had not planned to do so this year. "We believe that the North Korea approach is misguided in moving forward with its nuclear program to endanger peace and security in its homeland and neighbors in Asia," said Bolton, adding that the US supplies its nuclear weapons. "We don't want them to sell their nuclear weapons to other nations," he said. Bolton said the US would look for pressure on North Korea, which has been known to use nuclear weapons, as leverage to negotiations with the US. "I will reiterate what I have said before. So, the US has to put pressure on North Korea. But who else is going to hold the cards? Somebody else has to hold the cards," Mr Bolton said. Bolton described what the United States is prepared to do to get North Korea to agree to give up its weapons and asks for sanctions. "As far as I know, we have to use the pressure the reason for putting sanctions on North Korea," he said, adding that the US does not plan to ask the UN Security Council for sanctions alone.

> The defender is available for the Maribor first leg but his club believed he should be suspended. SNS Group Celtic made an administrative blunder which saw Efe Ambrose left behind

in the midfield in the Maribor department and has given him a potential three-match ban today. Although Efe Ambrose will be suspended next Friday, according to reports in Scottish media, the Celtic defender will still be fit for the Champions League first leg at Celtic Stadium in the middle of August. However, the Celtic club wrongly thought that Efe should only receive a three-match ban because he is available for the first leg. Although Efe Ambrose may receive a three-match ban next Friday, Efe Ambrose was part of the Celtic squad for the last match against Liverpool. However, says SNS Group Celtic he was making a tactical error and was left behind in midfield. It is understood that Efe Ambrose did not make the final squad and only played 11 games for the club this season. Efe Ambrose made his professional debut for Celtic in 2008 but spent nine months looking for a new club to return to. With a career-high 72 Celtic appearances, Efe is among Celtic's most capped players ever.

Carl Jara aka Grain Damaged is an award-winning, professional sand sculptor from Cleveland, Ohio. Jara says he has known since high-school that he wanted to be an artist. After studying

English and Art History at the Northeastern University, Jara says one semester he started carving a custom sculpture into sand molds, but didn't know how to do it. With the help of an instructor, he found out and learned how to use rubber molds to make art. Later, he made the decision to learn how to use sand and sculpt himself. In addition to how he makes his own sculptures, Jara says he does special events for comics companies such as Stan Lee and also for institutions like local community colleges and universities. In November of this year, he won the WWHS, The Very Art Of The Very Things Cleveland competition. Afterward, he will continue carving for clients in the comics industry and looks forward to sand sculpting in Cleveland. The Artist is professional sculptor who has been making art, for over 25 years, in various shapes and sizes.The artist says art is all about relationships and the best way to go into the heart is to create art. The artist has taught in various high schools in the Cleveland area and has taught a full time Honors Studio for High School students in grades 6, 7, and 8 time for over 20 years. Since Art is a personal form of artistic expression, he works individually in a way that allows the student that his work engages their imagination and presents ideas in ways that inform and challenge their own paths. Miguel Romano is a professional artist who worked in 3D modeling and animation in the areas of web design and production. The artist currently works as a digital artist in the ad and mass communication industries. In coming to Concrete Cleveland, he is excited to apply the 3D development and production skills he have to his work. The artist has a BFA in sculpture from Ohio University, along with an MFA in sculpture. We look forward to seeing his work very soon! Designed and installed by Concrete This Week. He is a guy originally from Cleveland, Ohio where he pursued a career as a nurse. He then moved to the Atlanta, GA area where he returned to school with a BSN and a BS in nursing and is a licensed nurse. He is a proud sorority brother and still has extra-curricular, as well as taking music lessons and the occasional play. He is a lovely asset at Concrete Cleveland and looks forward to seeing concrete

## I.2 Unconditional generation

Here's how that strategy works for your job:
1) You now plan upon what you accomplish to fulfill your goals.
Management cannot plan what happens to you. This may not be your ultimate personal decision, but it's perfectly fine to look at it. You just need to make sure you want to achieve this.
Now, because you've worked at goal, you don't have to talk about your status tomorrow, after all, you have to do your job and take care of yourself.
Next steps, there may be some work to do. There is a company down the road you right – literally millions of things that would have to be done. But of course it would have a different outlook. If you're going to do something, the customer might not be able to tell you.
2) Between those two steps are the plan in step so that your actions will be executed.
Then you have taken other steps (usually a few less important changes), like delivery. If you already know what that means, and you're having to stay up and take action you can make sure you don't have to point out in the moment to plan the action.
With business goals, it is not easy to pick up what appears best for us. We have to see what really is. What we do. We can't make a plan on the floor and come back up with exactly what you're doing. If you want to work every step, then you need to differentiate from the action and what the next step represents.
Eventually, you'll be less motivated to focus on this step and the previous one. Unfortunately if you don't change your main thing, you may be able to lose your motivation to work on "pivot." Unless that's possible, and if you don't change something, then the task may not be at the right time. Instead of doing something, it is just in advance of your ultimate goal.
Although you might make a mistake with every single day to day plan, it still is a great opportunity to correct your mistake, become new and commit to working extra hours and meeting your goals promptly.
The truth is, everything goes right for you no matter how quick a decision you become. The customer will never allow you to make the worst decisions. Otherwise, you make the very first decision.
3) Take timing as part of action. If you don't feel like you can keep it, a plan without help of timing stops you from doing. When it's like your plan in action can lead to something such as this: Now that you know what to do. For example, you might live in a place in the building that serves every customer, has 3 employees per team, and 3 clients on one. You will get things done the next day. Change your performance is the first step towards greater success, for example. Your team, at this point in the Customer department, will know how the customer deal with a single employee, the level chain, and more. Make sure you take action now that you change it. As a company, it won't be hard but you will have lots of work to do when you change.
4) Make sure it's your night.

Watch the humour but also the humanity behind the work we're doing. The truth is something very tragic and delicate in the middle of a very fractured world. It's the one thing that makes me proud. I feel like a singular individual has had to come together with this story. There's a lot of people who I've worked with for the very beginning, because I have got people, you've got people who have just had these eyes on this story, and this sense of what we are, that run through our final movie.
It's the very beginning we're at. The very beginning, we're not there, we will get there but we won't need. This is a story and these amazing actors, these fantastic violence, violence, that was just are elements and a complex world of conflict. When something like that is set to build this narrative and you're directing the world of these characters based around their individual needs it's very, extremely confusing, very heartbreaking — it's really quite intense—this was all built within it, and what it is, it's a 35-year old period that was slavery and still was very strong, these guys were operating to the edge and going to the point where we ended up setting up a big narrative, OK, that's good, it's okay, in some ways heroism is a noble imperative that we are fighting against, and recognize maturity as the mercurial nature and these are all human and we've got to clean it up so that that stuff is there and we've got to restore it. And the project we're looking at here is our common goal is that anything can be done to make that happen and everybody can do whatever their want to do and do it as they please.
That's the spirit of it. That's the movie I've made with Steven Wright in writing.

