# OpenReview forum: "Discrete Flow Matching"
_NeurIPS.cc/2024/Conference — NeurIPS 2024 spotlight_

### Official Review · Reviewer_7VEG · 2024-06-30

**Soundness:** 2
**Presentation:** 2
**Contribution:** 3
**Rating:** 8
**Confidence:** 4

**Summary:**

This paper presents Discrete Flow Matching, a new method for generating discrete data, such as language. The approach uses a general family of probability paths between source and target distributions. It offers a formula for sampling from these paths using learned posteriors. By focusing on specific probability paths, it improves generative perplexity compared to previous similar models. When scaled up, the method achieves notable performance on benchmarks. This approach bridges the gap between autoregressive models and discrete flow models.

**Strengths:**

The method is well-motivated for solving the discrete-state data in flow matching.

**Weaknesses:**

- In line 32, you mention one advantage of FM is its flexibility in handling non-Gaussian target distributions. Have you demonstrated this case?
- The method's description is unclear and confusing. I cannot distinguish which part pertains to Campbell's methods and which part is yours.
- In Table 2, it's unclear why the results from Austin et al. [2021a] are poor, as no explanation is provided.
- In Line 214, I'm not entirely sure what the difference is. Perhaps a table could clarify it?
- In Figure 3, the Inception Score (IS) needs to be included to demonstrate the diversity of image generation.
- Except the perplexity, how about the performance on BLEU, bertscore?
- Diversity is a significant concern for flow-based models. What about the result on diversity-related metrics?
- In F.1, in the context of conditional generation, it's unclear what the source (src) and target (tgt) samples are. Are they conditional prompts, target prompts, or a combination of Gaussian noise and target prompts?
- In section H, there is no qualitative analysis of unconditional generation.
- In Section H, I did not see the color red. I only observed the background colors: grey and yellow.
- It will be great that the code can be provided.
- Related works are missing.  it's important to note that flow matching has been utilized in various domains to capture the reader's interest. E.g., boosting diffusion[1],image generation[5], depth estimation[2], motiom[3], even text generation[4].

[1]. Boosting Latent Diffusion with Flow Matching

[2]. DepthFM: Fast Monocular Depth Estimation with Flow Matching

[3]. Motion Flow Matching for Human Motion Synthesis and Editing

[4]. Flow Matching for Conditional Text Generation in a Few Sampling Steps

[5], Latent Space Editing in Transformer-based Flow Matching

**Questions:**

as above

**Limitations:**

as above

---

> ### Author Rebuttal · Authors · 2024-08-07
>
> **Question: In line 32, you mention one advantage of FM is its flexibility in handling non-Gaussian target distributions. Have you demonstrated this case?** Yes, in fact all the probability paths we use in this paper are non-Gaussian. We worked with mask source distribution which corresponds to a delta function concentrated on a special “mask” token.
>
>
> **Comment: The method's description is unclear and confusing. I cannot distinguish which part pertains to Campbell's methods and which part is yours.** Thank you for the comment. We build upon Campbell’s work [1] but generalize it in ways that allow significant improvement in performance. In particular, we offer the following contributions: (i) we consider arbitrary data coupling $(X_0,X_1)$ and use it for conditioning; (ii) we offer a novel family of probability paths (equation 8) that generalizes the paths used in Campbell as particular cases; (iii) in particular, we show that incorporating polynomial schedulers $\kappa_t$ considerably improves performance; (iv) we provide a unified and closed-form formula for marginal probability velocity (rate) in equations 14, 16, 17 and show it has the exact same form as in continuous Flow Matching, see Table 1. Campbell provided this rate as an expectation and resorted to compute it individually for the masked and uniform noise cases; (v) we develop a general yet closed form formula for corrector sampling with arbitrary schedulers (equation 23). This generalizes Campbell’s stochastic sampling constant $\eta$ ($\alpha_t = 1 + t\eta$ and $\beta_t = \alpha_t - 1$), and we also note that Campbell’s stochastic sampling (Proposition 3.3 and equation 9 in [1]) incorporates the detailed balanced matrix in an implicit way and therefore requires a particular solution in each case; (vi) we show that particular polynomial correctors schedulers provide a further significant boost in results.
>
> [1] Andrew Campbell, Jason Yim, Regina Barzilay, Tom Rainforth, and Tommi Jaakkola. "Generative flows on discrete state-spaces: Enabling multimodal flows with applications to protein co-design."
>
> **Question: Why are the results from Austin et al. [2021a] poor?** We use the Pytroch implementation from https://github.com/cloneofsimo/d3pm, with the same architecture (DiT), tokenizer (GPT2), and data (Open Web Text) as we use for our model. We will add these details to the revised version.
>
> **Comment: In Line 214, I'm not entirely sure what the difference is.** As mentioned above (see (v),(vi) in the above answer) compared to [1] we provide more general (arbitrary schedulers) and closed form corrector steps. The discrete and continuous diffusion works [2, 3] define only corrector *iterations* and not sampling  (i.e., do not progress in time, similar to case (ii) described in line 280 in our submission) and perform corrector iterations by incorporating equal forward and reverse rates for the diffusion probability paths they consider. In our case we develop a closed form corrector step incorporating arbitrary schedulers (see $\alpha_t,\beta_t$ in equation (23)) that includes both corrector **sampling** and corrector **iterations** as special cases.
>
> [2] Andrew Campbell, Joe Benton, Valentin De Bortoli, Thomas Rainforth, George Deligiannidis, and Arnaud Doucet. "A continuous time framework for discrete denoising models."
>
> [3] Yang Song, Jascha Sohl-Dickstein, Diederik P Kingma, Abhishek Kumar, Stefano Ermon, and Ben Poole. “Score-based generative modeling through stochastic differential equations.”
>
>
> **Comment: In Figure 3, the Inception Score (IS) needs to be included to demonstrate the diversity of image generation.** Per the reviewer’s request, we added a corresponding inception score graph in the PDF attached to the main response of rebuttal. Note that we observed a similar trend to the FID demonstrated in the original submission.
>
> **Question: Except the perplexity, how about the performance on BLEU, bertscore?** BLEU and BERT-score are traditionally used for the “data-data” case (e.g., translation), that is when both source and target samples are data, and then generated samples are compared to target (test) data. We use “noise-data” (except the conditioning part) and therefore we couldn't find a reasonable way to apply these metrics in our case.
>
> **Comment: Diversity is a significant concern for flow-based models. What about the result on diversity-related metrics?** We are not aware that flow-based models have a diversity concern in general, maybe the reviewer refers to distilled flow models?  In any case, we didn’t encounter particular diversity issues with our model, and in Tables 2, 3, and Figure 7, with additional details provided in Appendix F.1, we present the entropy of tokens within generated sequences to illustrate the diversity of the model's predictions. Furthermore, our modeling allows for control over the diversity by adjusting the temperature during sampling from  $p_{1|t}$.
>
> To further address the reviewer’s concerns, we attach in the rebuttal’s main reply several uncurated generations, sampled with the same prompt. These uncurated generations demonstrate the diversity of our model's output.
>
> **Comment: In F.1, in the context of conditional generation, it's unclear what the source (src) and target (tgt) samples are. Are they conditional prompts, target prompts, or a combination of Gaussian noise and target prompts?** In the context of conditional generation (Appendix F.1), the source-target pairs $(X_0,X_1)$ are as described in Equation 5 in the paper, i.e., $(X_0,X_1) = (\mathbb{I} \odot X_1  + (\mathbf{1}-\mathbb{I})\odot (\mathbb{m},\ldots,\mathbb{m}) , X_1)$, where $\mathbb{I} \in \set{0,1}^{N}$ is indicating the conditioning mask. There are no target prompts, nor a combination of Gaussian noise/target prompts. We did our best to understand the reviewer’s question, if we got it wrong, please clarify.

---

> ### Author Response · Authors · 2024-08-07
>
> **Comment: In section H, there is no qualitative analysis of unconditional generation.** Per the reviewer's suggestion, we added unconditional samples, generated by our model (attached in the pdf in the main response). We will add these samples to the revised version.
>
> **Commnet: In Section H, I did not see the color red. I only observed the background colors: gray and yellow.** This is a mistake, prompts are marked in gray, we will fix it in the revised version.
>
> **Comment: It will be great that the code can be provided.** We are planning on releasing the code not much after the paper is published.
>
> **Comment: Related works are missing.** We thank the reviewer for the suggestion, we will add the relevant related works to the revised version.

---

> > ### Comment · Reviewer_7VEG · 2024-08-07
> > **thanks for your reply**
> >
> > i am looking forwards to the code.

---

> > > ### Author Response · Authors · 2024-08-11
> > >
> > > We are planning to release the code around the time of publication. As we near the end of the discussion period, we would be happy to address any further clarifications or concerns regarding the other comments raised by the reviewer.

---

> > > > ### Comment · Reviewer_7VEG · 2024-08-13
> > > > **strong accept**
> > > >
> > > > Thanks for the author's reply; my concerns are resolved.
> > > >
> > > > I have one suggestion for the draft: emphasize more in the main paper about the instantiation of the scheduler, as it's critical for performance.

---

### Official Review · Reviewer_vMd5 · 2024-07-12

**Soundness:** 4
**Presentation:** 3
**Contribution:** 3
**Rating:** 7
**Confidence:** 4

**Summary:**

The paper proposes an approach to generative modelling for discrete data, i.e. multidimensional distributions where the variable along every dimension can take value in a finite set. This is an alternative approach to autoregressive generative modelling for discrete data which is currently actively studied for language and code generation.

The philosophy of the proposed method is heavily inspired by the Flow Matching algorithm [1] and Continuous Time Markov Chains (CTMC), which were previously used in [2] to propose a similar model. That is, for a given CTMC, the authors define the vector field generating samples from this CTMC by local updates of the samples (independently along every dimension). Based on the PMF (Probability Mass Function) of the CTMC, the authors derive a formula for the vector field. Furthermore, they introduce the continuity equation analogous to the continuous case, which allows for an easy validation that the change of the density given by CTMC corresponds to the vector field.

The authors perform an ablation study of some of their design choices and extensive empirical studies for generation of code, language, and discrete-valued images. The proposed model outperforms the competitors bridging the gap between flow-based models and autoregressive models.

[1] Lipman, Yaron, Ricky TQ Chen, Heli Ben-Hamu, Maximilian Nickel, and Matt Le. "Flow matching for generative modeling." *arXiv preprint arXiv:2210.02747* (2022).

[2] Campbell, Andrew, Jason Yim, Regina Barzilay, Tom Rainforth, and Tommi Jaakkola. "Generative flows on discrete state-spaces: Enabling multimodal flows with applications to protein co-design." arXiv preprint arXiv:2402.04997 (2024).

**Strengths:**

The paper presents a complete study of the actively studied topic in the field. It is clearly written, the developments presented in the paper are novel and are properly studied empirically. The paper is of great interest to the NeurIPS community.

**Weaknesses:**

The paper presents a complete study and its methodological part does not raise any major concerns. However, I would like to ask the authors to clarify the following question.

Why do posteriors in eq. 15 define a proper distribution? Indeed, according to eq. 8, the linear combination of the conditional distributions with the schedulers should define a correct distribution (sum up to $1$). If we simply try to sum over all possible $x^i$ in eq. 15, we won’t get $1$, hence eq. 15 does not define a posterior. This is an important detail when defining the objective in eq. 25, which is the cross entropy between distributions.

The minor

- Descriptions of the schedulers on lines 96 and 98 are not complete, i.e. there are no conditions that schedulers have to define correct probability distributions in eqs. (9,10) and the scheduler $\kappa^3$ is missing a description.
- Figure 2. The plots presented in the figure have $d = 2$ unlike what is stated in the caption.
- At the beginning of the paper, the authors introduce the notation of random variables as capital letters (e.g. $X_t$). This creates confusion in equations 13-17 because there the authors clearly mean $X_t$ to be the value of a random variable.
- There is a typo in line 160 when describing the panel of Fig. 2.
- There is a typo in line 163 when defining the vector field $v(x,z)$.
- There is a type in line 582. Index $\ell$ is not a function of $j$ according to eq. (36).
- I haven’t checked thoroughly, but I think equations on the top of page 19 should have a summation over $j\neq \ell$ instead of the summation over all possible $j$.

**Questions:**

I would suggest adding a discussion of the variable-length generation. It is an important difference with autoregressive modelling and I’m wondering how the authors handle this issue given that their model has two important properties:

1. Conditional generation for partially masked sequences.
2. Independence of the conditional distributions between dimensions.

The questions I would like to have answered are:

1. Does one have to define the length of the generated sequence before generation? If so how this can be decided?
2. What’s the computational cost if one continues generating the sequence in the autoregressive way?

**Limitations:**

NeurIPS Paper Checklist does not follow the required format.

The paper adequately discusses the limitations of the proposed approach. The only thing that is not cover enough in the paper is the variable-length generation.

---

> ### Author Rebuttal · Authors · 2024-08-07
>
> **Question: Why do posteriors in Equation 15 define a proper distribution?** Equation 15 is a proper distribution as follows:
> $$\sum_{x^i} \hat{w}^j_t(x^i|X_t) = \sum_{x_0,x_1} \overbrace{\big( \sum_{x^i} w^j(x^i|x_0,x_1)\big )}^{=1} p_t(x_0,x_1|X_t) = \sum_{x_0,x_1} p_t(x_0,x_1|X_t) = 1,$$
> where in the first equality we change the summation order.
>
> **Comment: Descriptions of the schedulers on lines 96 and 98 are not complete.** The conditions in lines 96 and 98 should be understood as _additional conditions_ to the general conditions presented in  Line 91, i.e., $\sum_{j} \kappa_{t}^{i,j}=1$ and $\kappa_{t}^{i,j} \ge 0$. Combining the conditions in Line 91 with the ones in Lines 96 and 98 guarantees proper distributions. We realize now this is confusing and will clarify this in the revised version of the paper - thank you.
>
> **Comment: Figure 2. The plots presented in the figure have $d=2$ unlike what is stated in the caption.** This is indeed a typo, but should be $d=4$ (and $N=2$), that is, the figure depicts the state space of two tokens $x=(x^1,x^2)$ where each token can get a value in a vocabulary of size $d=4$.
>
> **Comment: Random variable notation creates confusion.** We agree with the reviewer that this notation is confusing and we will use a lower case letter, e.g., $z$, instead in the revised paper.
>
> **Comments: Typos Lines 160, 163, and 582.** Thanks, we will fix it in the revised version.
>
> **Comment: Summation on page 19.** Please note that in the case of $j=\ell$ the term in the summation equals zero, thus, summation over $j \ne \ell$ will yield the same result.
>
> **Question: Does one have to define the length of the generated sequence before generation? If so how this can be decided?** No, the length does not have to be defined before generation. We train the model using flattened data samples which are separated by an end-of-text (EOT) token. The model can predict EOT token in any location in the generated sequence $x^1,x^2,\ldots,x^N$ and the EOT will indicate the end of the generated text (similar to autoregressive modeling).
>
> **Question: What is the computational cost if one continues generating the sequence in the autoregressive way?** The model generates a sequence of length $\leq N$, where $<N$ length can happen if the EOT token is predicted. If it does not (some texts are longer than $N$ tokens) one can potentially continue to generate by conditioning on the last $K<N$ tokens and predict the next $N-K$ tokens, where $K$ is user defined: e.g., large $K$ will provide more context but shorter extension. The cost of predicting these $N-K$ new tokens will be equivalent to a full sequence generation with our model. It is very interesting to develop methods to accelerate the generation, but we leave it to future research.

---

> > ### Comment · Reviewer_vMd5 · 2024-08-07
> > **response acknowledgment**
> >
> > Thank you for your response! Everything is clear. Sorry for the confusion in some of the questions.

---

### Official Review · Reviewer_ppDj · 2024-07-13

**Soundness:** 4
**Presentation:** 4
**Contribution:** 3
**Rating:** 6
**Confidence:** 3

**Summary:**

The paper introduces a novel approach called Discrete Flow Matching, which adapts continuous flow models to discrete sequential data. It extends prior work by integrating discrete state spaces and time-dependent schedulers into a unified framework for non-autoregressive generative modeling. Methodologically, Discrete FM employs generating probability velocities derived from learned posteriors and schedulers, enabling efficient sampling and correction processes. This approach facilitates the transformation of noise distributions into target data distributions with enhanced flexibility and performance. Experimental evaluations across language modeling, code generation, and image generation tasks demonstrate significant improvements over existing methods. Specifically, the model achieves state-of-the-art results on various benchmarks, including HumanEval and MBPP coding tasks, showcasing its efficacy in generating high-quality outputs.

**Strengths:**

- The methodology design is solid with formal analysis and mathematical proofs.

- The paper conducted a set of experiments in language modeling, code generation, and image generation tasks, showcasing the most promising results to date in a non-autoregressive context.

- Technical details about the methodologies are rich. And the experimental setup includes detailed descriptions of the methodologies used, such as masked source training, conditional couplings, probability path schedulers, and corrector steps, which are pivotal for the model's performance.

**Weaknesses:**

- The method is heavily based on the Continuous-Time discrete Markov Chain (CTMC) paradigm from Campbell et al., although this paper proposed theoretical and empirical improvements such as the unified formulation for more general probabilistic paths and velocity as well as the scheduler designs.

- There are still performance gap in code generation evaluation when compared with autoregressive language models.

**Questions:**

- I am a bit confused with Eq 10 especially the scheduler terms. Please explain or tell if there are mistakes.

- What are the main reasons for discrete flow matching to require significantly large number of evaluations?

**Limitations:**

The authors acknowledged their limitations such as the number of evaluation is high compared to continuous flow matching.

---

> ### Author Rebuttal · Authors · 2024-08-07
>
> **Comment: I am a bit confused with Eq 10 especially the scheduler terms.** Equation 10 proposes a second  instantiation of a conditional probability path $p_t(\cdot|x_0,x_1)$ where given some pair $(x^i_0,x^i_1)$ of source and target tokens, the token at time $t$ is: $x^i_1$ with probability $\kappa^1_t$; $x^i_0$ with probability $\kappa^3_t$; and a uniformly distributed random token with probability $\kappa^2_t$. This probability path is reminiscent of the brownian bridge in the continuous diffusion world. Lastly, note that the conditions on the schedulers $\kappa^j_t$ that guarantee this conditional path interpolates between $\delta_{x_0}(x^i)$ and $\delta_{x_1}(x^i)$  and stays a proper PMF for all $t\in[0,1]$ are the ones described in lines 98 and 91. We will clarify this in the revised version.
>
>
> **Question: What are the main reasons for discrete flow matching to require a significantly large number of evaluations?** Discrete flow matching requires a higher number of function evaluations compared to its (deterministic) continuous counterpart. We attribute this to its stochastic sampling, similar to sampling by approximating a solution of an SDE which typically possesses a lower strong convergence order than their ODE solvers counterparts. In particular, for the Euler sampling case the deterministic Euler method has strong convergence order of $1$ while the non-deterministic Euler (Euler-Maruyama) has strong convergence order of $\frac{1}{2}$, see e.g., [1]. While this gives some intuition we do agree an analysis of global convergence of discrete sampling is interesting and defer it to future work; in this paper we only show local convergence error (see $o(h)=O(h^2)$ term in equation (22)).
>
> [1] Sauer, T., 2011. Numerical solution of stochastic differential equations in finance. In Handbook of computational finance (pp. 529-550). Berlin, Heidelberg: Springer Berlin Heidelberg.

---

> > ### Comment · Reviewer_ppDj · 2024-08-12
> >
> > Thanks for the response that addressed my concerns. I would keep the rating to recommend acceptance.

---

### Official Review · Reviewer_oYBb · 2024-07-13

**Soundness:** 3
**Presentation:** 3
**Contribution:** 3
**Rating:** 5
**Confidence:** 4

**Summary:**

The paper presents a discrete flow matching method for modeling discrete data with discrete state space. The paper presents unified frameworks for training and sampling from the discrete probabilistic model. Importantly, the paper also studied scaling up the model to 1.7B parameters and tested the model on code generation tasks. Comprehensive experiments shows that the proposed method outperform existing methods and also close the gap to autoregressive models.

**Strengths:**

1. Impressive empirical studies with 1.8B scaling up model. To my knowledge, this is the first discrete diffusion that scaled up to this scale.

2. The paper provides a principled view of the discrete diffusion model, and proposes several novel techniques such as backward sampling in conditional generation scenarios.

3. The paper is very clearly written and well organized.

**Weaknesses:**

The paper lacks some experiments on common benchmarks used in existing discrete diffusion models, e.g., LM1B and OWT. I understand the scaling-up experiments on HumanEval are more challenging, but I believe a comparison on the common benchmark would be important to justify the effectiveness over existing methods.

**Questions:**

See weakness.

**Limitations:**

I didn't find limitations specifically discussed in the paper, though the author marked it discussed in the checklist. Please correct me if I missed it.

---

> ### Author Rebuttal · Authors · 2024-08-07
>
> **Comment: The paper lacks some experiments on common benchmarks used in existing discrete diffusion models, e.g., LM1B and OWT.** First, please note that our experimental setup already includes the OWT dataset (see Table 2 and lines 268-273). Second, per the reviewer’s suggestion, we trained and evaluated our and baselines models on the LM1B dataset, please see Table 1 in the main rebuttal reply.
>
> _Experimental setup._ For the reviewer’s convenience, we would like to detail all the evaluations done in this work. We experiment with two data modalities, text and images on small and large scale datasets.
>
> | Modality | Scale | Datasets | Compared to | Metrics | Comments |
> |---|---|---|---|---|---|
> | Text | Small | Open web text (OWT) | State-of-the-art prior works, autoregressive modeling | Generative perplexity (using Llama 2, 3, and GPT2), Entropy, NFE | All models are evaluated without temperature annealing.  |
> | Image | Small | CIFAR10 | Campbel et al., Maskgit | FID, added inception score in this rebuttal, NFE |  |
> | Code | Large | Large scale code mix | Autoregressive (note: no other discrete diffusion/flow worked on this tasks before) | Pass@1, Pass@10, Pass@25 on Humaneval and MBPP |  |
> | Text | Large | Large scale text mix | Autoregressive, Savinov et al. | Generative perplexity (using Llama 2, 3, and GPT2), Entropy, NFE |  |
>
> **Comment: I didn't find limitations specifically discussed in the paper.** In the conclusions (Section 5), we mention the following limitations: (1) Discrete flow matching requires a higher number of function evaluations compared to its (deterministic) continuous counterpart. We attribute this to its stochastic sampling, similar to sampling by approximating a solution of an SDE which typically possesses a lower convergence order than their ODE solvers counterparts (see e.g., [1]). (2) There remains a performance gap between autoregressive modeling and our proposed approach.
>
> [1] Sauer, T., 2011. Numerical solution of stochastic differential equations in finance. In Handbook of computational finance (pp. 529-550). Berlin, Heidelberg: Springer Berlin Heidelberg.

---

### Author Rebuttal · Authors · 2024-08-07

We appreciate the reviewers' insightful feedback on our paper. We address each of the comments/questions raised by the reviewers in the specific threads below. We would be happy to address any remaining concerns during the discussion period.

Here we summarize the new experiments we performed during rebuttal period to address remaining concerns of reviewers:
1. As suggested by reviewer oYBb, in Table 1, we report a new comparison to the baselines on the LM1B dataset.
| Method | NFE | Llama-2 | Llama-3 | GPT2 | Entropy |
|---|---|---|---|---|---|
| Data |  | 5.9 | 7.7 | 17.6 | 8.0 |
| Han et al. | >10000 | 67.8 | 97.6 | 150.1 | 8.0 |
| Lou et al. | 256/512/1024 | 27.9/26.1/23.7 | 41.7/39.2/35.0 | 129.3/120.9/104.2 | 8.1/8.1/8.2 |
| Campbell et al. | 256/512/1024 | 26.4/25.7/23.1 | 39.7/34.9/32.6 | 121.7/115.1/99.3 | 8.2/8.2/8.2 |
| FM (Ours) | 256/512/1024 | **24.2/23.4/20.8** | **35.2/33.2/29.2** | **110.6/106.6/86.7** | 8.2/8.2/8.1 |
2. As suggested by reviewer 7VEG, we have computed the inception score for both our proposed and baseline CIFAR10 models. The results are presented in the attached PDF, and demonstrate a similar trend as with the reported FID metric in the submission.
3. As suggested by reviewer 7VEG, we added unconditional samples of our 1.7B Discrete Flow Matching text model to the rebuttal’s PDF.
4. To address reviewer 7VEG concerns about diversity of predictions, we added below more **uncurated** qualitative samples, generated with the same prompt.

**Uncurated qualitative conditional samples**

**Einstein was a German-born theoretical physicist who is widely held to be one of the greatest and most influential scientists of all time. Best known for developing the** famous theory of relativity, Einstein was also well known for his advocating for women in the field of science. Find out everything to know about him and scientific theories about the universe.
Albert Einstein is widely held to be one of the most significant scientists of the 21st century. He is also acknowledged for the multibillion impact of his work.
Described as a mathematician, physicist and philosopher, he is mainly regarded for his theories on the universe. And so, he is also considered to have had an immense impact. It is estimated that somewhere around 10,000 people contributed towards the creation of his theories. As a result, the concepts, that he created, and in his case, that of his brilliant genius, and his theories that afforded him the Theory of Relativity, have all consolidated until this day. So, let’s learn everything to know about Einstein and all of his theories and works.


**Albert Einstein was a German-born theoretical physicist who is widely held to be one of the greatest and most influential scientists of all time. Best known for developing the theory of the** earth movement, the theory of relativity, and the principles of the theory of general physics and the theory of special relativity, which are precursors to the quantum theory.
He was the second leading theoretical physicist in the 20 history of science and a cosmopolitan statesman. He is among the most important and influential figures of all time in the development of best known for developing the theory of the earth movement, the theory of relativity and the principles of the theory of general physics and the theory of special physics. Einstein's theories of relativity, which emerged in the first decade of the twentieth century, describe the relationship between the two poles of the universe. His theories also summarize the relationship between gravity and electromagnetism, the speed of light (sometimes used as the "speed"), special effects including telekinesis (the illusion of self), and time travel, and a popular but flawed quantum theory.


**Albert Einstein was a German-born theoretical physicist who is widely held to be one of the greatest and most influential scientists of all time. Best known for developing the** laws of motion and the theory of general relativity, Einstein's contributions developed over time to create his final achievement in physics, relativity with its general theories and theory of gravity, sometimes called the special theory of relativity. He is widely considered as the father of modern quantum mechanics, wrote his Aclonomical Revolution (one could be called the greatest pedagogues of all time), revealed the Newtonian Square of suggestion, the Doppler effect for light and a gyromagnetic effect, all of which lead to the standard physics of quantum mechanics.

---

### Author Response · Authors · 2024-08-11

As we approach the deadline for the authors-reviewers discussion, we would like to kindly inquire if there are any additional questions from the reviewers. We are happy to address any remaining clarifications or concerns.

---

### Decision · Program_Chairs · 2024-09-25

**Decision:**

Accept (spotlight)

**Comment:**

The paper introduces a novel variant of the flow-matching model for discrete data types. The authors provide solid theoretical foundations for their approach and conduct experiments in language modeling, code generation, and image generation tasks. The model can be seen as an alternative approach to autoregressive models and possibly opens new research directions in generative models for discrete data.

The reviewers are pretty consistent in evaluating the presented approach's quality and novelty, and many concerns were solved during the rebuttal phase. The idea of discrete flow matching was initially introduced by Campbell et al., 2024, which may slightly lower the contribution. However, the authors clearly point out the differences and compare quantitatively with this approach considering the image generation task. Therefore, the paper should be accepted.